# Amplifying Membership Exposure via Data Poisoning

**Yufei Chen**[1,2]   **Chao Shen**[1]   **Yun Shen**[3]   **Cong Wang**[2]   **Yang Zhang**[4]

[1]Xi'an Jiaotong University   [2]City University of Hong Kong
[3]NetApp   [4]CISPA Helmholtz Center for Information Security

## Abstract

As in-the-wild data are increasingly involved in the training stage, machine learning applications become more susceptible to data poisoning attacks. Such attacks typically lead to test-time accuracy degradation or controlled misprediction. In this paper, we investigate the third type of exploitation of data poisoning - increasing the risks of privacy leakage of benign training samples. To this end, we demonstrate a set of data poisoning attacks to amplify the membership exposure of the targeted class. We first propose a generic *dirty-label* attack for supervised classification algorithms. We then propose an optimization-based *clean-label* attack in the transfer learning scenario, whereby the poisoning samples are correctly labeled and look "natural" to evade human moderation. We extensively evaluate our attacks on computer vision benchmarks. Our results show that the proposed attacks can substantially increase the membership inference precision with minimum overall test-time model performance degradation. To mitigate the potential negative impacts of our attacks, we also investigate feasible countermeasures.[1]

## 1 Introduction

Training data are the most critical ingredients of machine learning, which are sometimes regarded as a new type of fuel in the era of artificial intelligence [30]. Over the last decade, training data collection and preservation have received growing concerns [10, 29], among which two focal agendas arise.

One is data corruption caused by *data poisoning* attacks, posing serious threats to training data **integrity**. Data poisoning attacks exploit the common practice to (usually automatically) collect training data in the wild, e.g., from the Internet. This practice opens doors for attackers to inject malicious data at the training time to manipulate model behaviors at the test time. Consequently, models trained with poisoned training data suffer from either accuracy degradation [7, 19], targeted misclassification [34, 55], or backdoor implantation [32, 50].

The other is data leakage caused by *privacy inference* attacks, mainly violating training data **confidentiality**. Ideally, a machine learning model learns generalizable knowledge of the training data, rather than details of specific data points. However, previous studies show that conventional learning algorithms can unintentionally remember sensitive information. Such information can later be inferred by various privacy inference attacks, such as membership inference attacks [33, 38]. Essentially, data poisoning is a training time attack, while privacy inference is performed at the testing time. Despite intensive research efforts, most existing works study them separately. It remains unclear whether these two attacks can be integrated to cause more severe damage to machine learning models.

In this paper, we aim to advance the research frontier on the connection between data integrity and confidentiality. To this end, we propose a set of poisoning attacks to increase the precision of membership inference attacks, which is widely adopted as a standard tool to measure privacy leakage in statistical analysis [15, 40]. In particular, we first show a simple but effective dirty-label poisoning

---

[1]Code is available at https://github.com/yfchen1994/poisoning_membership.

36th Conference on Neural Information Processing Systems (NeurIPS 2022).

attack that is generic for supervised classification applications. We then demonstrate a clean-label poisoning attack applicable to the transfer learning scenario. It has two significant advantages: (1) no requirements on the labeling process and (2) natural appearances to human moderation. Consequently, our proposed clean-label poisoning attack challenges the case where automatically crawling data from the Internet has become a common practice.

Our contributions are as follows: (1) We reveal an underexplored data poisoning attack threatening the training data privacy. In particular, we demonstrate how to amplify membership inference exposure of a specific class with only slight impacts on the model performance. (2) In generic supervised learning settings, we introduce a naïve dirty-label poisoning attack by modifying the labels of the poisoning samples. (3) In the transfer learning setting, we propose an optimization-based clean-label poisoning attack, wherein for each poisoning sample, we make imperceptible modifications without changing its label. It presents a more practical attack example since it does not require control over the labeling process. It is also a more stealthy attack as the contents of poisoning instances "look similar" to the natural ones. (4) We conduct extensive empirical studies to investigate the impacts of various factors, including datasets, architectures, poisoning budgets, and learning setups. Our results show that our proposed poisoning attacks are able to increase the membership exposure obviously with just limited influences on the victim model's performance on testing samples. (5) At the end, we also consider several potential countermeasures and investigate their effectiveness.

## 2  Background

In this work, we focus on deep supervised classification in the computer vision field. That is, given a training dataset $\mathcal{D}_{\text{train}}$ composed of input-label pairs $(x, y)$, a deep classifier $f(x; \theta)$ with model parameters $\theta$, and a learning objective $\mathcal{L}\left(f(x; \theta), y\right)$, the learning process aims to find a set of model parameters $\theta^*$ to minimize the learning objective on $\mathcal{D}_{\text{train}}$, i.e., $\theta^* = \arg\min_{\theta} \sum_{(x,y) \in \mathcal{D}_{\text{train}}} \mathcal{L}\left(f(x; \theta), y\right)$.

### 2.1  Data Poisoning

In a poisoning attack, an attacker crafts and injects a set of malicious training samples $\mathcal{D}_{\text{poison}}$ (i.e., *poisoning dataset*) into the benign dataset $\mathcal{D}_{\text{clean}}$ (i.e., *clean dataset*). In the training stage, the model holder executes the machine learning algorithm on the full *poisoned dataset* $\mathcal{D}_{\text{train}} = \mathcal{D}_{\text{clean}} \cup \mathcal{D}_{\text{poison}}$ to obtain the trained model $f(x; \theta^*)$. In the inference stage, $f(x; \theta^*)$ tends to show unexpected behaviors on targeted inputs $(x, y) \in \mathcal{D}_{\text{target}}$. Most existing literature poses the poisoning attack as a bi-level optimization problem [8, 16]:

$$
\begin{aligned}
\mathcal{D}_{\text{poison}} &= \arg\min_{\mathcal{D}} \sum_{(x,y) \in \mathcal{D}_{\text{target}}} \mathcal{A}\left(f(x; \theta^*)\right), \\
\text{s.t. } \theta^* &= \arg\min_{\theta} \sum_{(x,y) \in \mathcal{D}_{\text{clean}} \cup \mathcal{D}_{\text{poison}}} \mathcal{L}\left(f(x; \theta), y\right)
\end{aligned}
\tag{1}
$$

where $\mathcal{A}$ is the adversarial objective of the poisoning attack. Typically, there are three adversarial objectives: (1) in the accuracy degradation case, $\mathcal{D}_{\text{target}}$ contains all testing samples and $\mathcal{A} = -\mathcal{L}(f(x; \theta^*), y)$; (2) in the targeted misclassification case, $\mathcal{D}_{\text{target}}$ contains samples expected to be misclassified into the target class $t$ and $\mathcal{A} = \mathcal{L}(f(x; \theta^*), t)$; (3) in the backdoor implantation case, $\mathcal{D}_{\text{target}}$ involves samples with a trigger $\delta$, where inputs embedded with the trigger are to be classified into the target class $t$ and $\mathcal{A} = \mathcal{L}(f(x \oplus \delta; \theta^*), t)$.

Based on the attacker's capability, poisoning attacks can be further grouped into two categories:

- **Dirty-label Poisoning.** Most classical poisoning attacks require the control of the labeling process, where the attacker is allowed to modify the labels in $\mathcal{D}_{\text{poison}}$ [19, 42, 48]. Such an attack is called *dirty-label poisoning*. Despite promising attack performance, dirty-label poisoning becomes impractical in supervised machine learning scenarios for two main reasons. First, in reality, it is a common practice that only unlabeled data (e.g., images and videos) are scraped by crawlers and then labeled by human moderators. Second, machine learning developers usually utilize anomaly detection algorithms to filter out wrongly labeled

samples. Therefore, there exists a minimal chance that the data samples with modified labels are preserved in the dataset.

- **Clean-label Poisoning.** To overcome the shortcomings of dirty-label poisoning, recent studies propose *clean-label poisoning* attacks [34, 53, 55]. A clean-label poisoning attack satisfies two properties. First, the labels of poisoning samples remain unchanged in the poisoning process. Second, to be inconspicuous, each poisoning sample $x_p$ visually resembles a natural sample $x_n$, which is always constrained by a $L_p$-norm distance $\|x_p - x_n\|_p < \epsilon$.

**Our Setup.** In this work, we investigate data poisoning under a newly identified and underexplored adversarial objective: *privacy leakage amplification*. In particular, we focus on one of the most representative privacy inference attacks: membership inference attack [33, 38]. We start from the dirty-label poisoning setup, which establishes a baseline and cornerstone for more advanced attacks. Then, we investigate attack methods in the clean-label poisoning setup to improve practicability. Note that there are concurrent works that propose to use poisoning attacks to cause privacy leakage [26, 43].

## 2.2 Membership Inference Attack

In a membership inference attack, an attacker aims to infer whether a specific sample $(x, y)$ belongs to the training dataset $\mathcal{D}_{\text{train}}$ at the test time [24, 33, 38]. Unintended membership exposure causes catastrophic privacy loss for individuals. For example, in the real world, a data sample $x$ can be a clinical record or an individual. Membership inference attacks enable the attackers to infer whether this clinical record or individual has been used to train a model associated with a certain disease. As such, these attacks are widely adopted as basic metrics to quantify privacy exposure in statistical data analysis algorithms [13, 23]. Henceforth, we use membership inference attacks to demonstrate how to exploit data poisoning to amplify privacy leakage in this paper.

Based on the attacker's capability, membership inference attacks can be grouped into two categories:

- **Black-box Membership Inference.** In this case, the attacker distinguishes members and non-members only using model outputs [17, 37, 38]. This case is generic to most machine learning contexts. There are two attack strategies in general. The first strategy is *model-based* [38], where the attacker builds multiple shadow models to mimic the victim model, then utilizes them to construct a dataset with member/non-member labels, and finally trains a binary classifier to predict member/non-member. The second strategy is *metric-based* [33, 39], where the attacker compares a designed metric $M_{\text{mem}}$, such as correctness or entropy, with a predefined threshold $\tau$ to infer if a sample belongs to the training dataset.

- **White-box Membership Inference.** In this case, model parameters $\theta^*$ or even intermediate training information such as gradients $\frac{\partial \mathcal{L}}{\partial \theta}$ are observable by the attacker [22, 27, 31]. Such capability provides additional information supporting inference attacks, which is usually achievable in collaborative learning settings. Most white-box attacks are model-based, as the attacker needs to access the internals of deep models to extract model-specific features.

**Our Setup.** Our evaluation is carried out in the metric-based black-box membership inference setting. We adopt the following metric proposed by Song et al. [40] to measure the membership exposure:[2]

$$M_{\text{mem}} = -\left(1 - f(x)_y\right) \log\left(f(x)_y\right) - \sum_{i \neq y} \log\left(1 - f(x)_i\right) f(x)_i \tag{2}$$

where $f(x)_j$ refers to the confidence value of label $j$. Equation 2 simultaneously considers the correctness and entropy metric. In the training stage, the learning algorithm continuously fits the training samples, by decreasing the entropy loss (i.e., the learning objective) and increasing the confidence score of the correct label. As a result, a member is likely to produce a lower $M_{\text{mem}}$ than a non-member. To gain a holistic view of the membership exposure, we calculate the true positive rate (TPR) and false positive rate (FPR) of the membership inference attack by varying the threshold $\tau$, and plot the ROC curve. Then we adopt the AUC (area under the ROC curve) score to measure membership exposure. In general, a higher AUC score means a higher risk of membership exposure, as we can find a threshold $\tau$ with high TPR and low FPR. It is notable that the AUC score is an average-case metric, which hardly effectively reflects the worst-case privacy [9]. Therefore, we also

---

[2]For simplicity, we use the notation $f(x)$ to represent the trained model $f(x; \theta^*)$.

report the TPR when the FPR is low (1% in our case), which works as a compensation for the AUC metric to indicate the worst-case privacy.[3]

## 2.3 Threat Model

Before diving into our attack design, we first elaborate on the threat model considered in this paper.

**Attack Goals.** The first goal of the attacker is *increasing the chance of leaking the membership of training samples within a targeted class*. At the same time, the attacker attempts to make the poisoning attack as stealthy as possible. As such, the second goal of the attacker is *generating poisoning samples that have limited impacts on the model performance for untargeted classes and are indistinguishable from natural samples*.

**Attacker Capabilities.** We assume the attacker has the basic capabilities set up by existing data poisoning and membership inference attack games. The attacker can craft poisoning samples and inject them into the victim's clean dataset $\mathcal{D}_{\text{clean}}$. However, there exists a *poisoning budget* $b_{\text{poison}}$ to limit the amount of training samples ($|\mathcal{D}_{\text{poison}}| \leq b_{\text{poison}} \ll |\mathcal{D}_{\text{clean}}|$). We assume the attacker owns a *shadow dataset* $\mathcal{D}_{\text{shadow}}$ to craft poisoning samples, which contains natural samples from the same distribution with $\mathcal{D}_{\text{clean}}$. Moreover, the attacker cannot modify poisoning data labels in the clean-label poisoning setting. But we assume the attacker knows the feature extractor (i.e., an encoder) used by the victim model. This assumption is practical since developing models with public pretrained feature extractors is a common practice in existing clean-label poisoning literature [34, 53].

**Remark.** In our setup, the clean dataset $\mathcal{D}_{\text{clean}}$ is unknown by the attacker, which is different from most existing data poisoning games [7, 19, 26].

## 3 Attack Methodology

Although Equation 1 establishes a generic framework for poisoning attack, it is unsuitable for our case. First, classic solutions for the optimization problem posed by Equation 1 require differentiation to the inner loss minimizer, which is intractable to models with high complexity. Second, although meta-learning algorithms have been proposed to solve the bi-level optimization problem in the deep learning setting [16, 52, 54], they lead to huge computing costs, limiting the practicability of the attack. Third, in our threat model, $\mathcal{D}_{\text{clean}}$ is unobservable, making Equation 1 no longer applicable to our attack. Facing these obstacles, we explore heuristic strategies to achieve the attack goals.

### 3.1 Dirty-label Poisoning Attack: A Naïve Approach

We start our investigation in the dirty-label poisoning setting. This case has more limitations in the real world as discussed in Section 2.1. But it relaxes the attack constraints and helps to verify the feasibility of our attack strategy.

The key to amplifying membership exposure is to cause overfitting in the targeted class. We propose the following attack strategy:

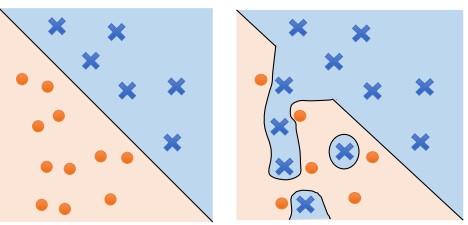

**Clean Model**     **Dirty-label Poisoned Model**

Figure 1: Conceptual illustration of the decision boundary of models trained without/with our dirty-label poisoning attack. To achieve a low training accuracy, the model has to learn a more complicated decision boundary, making the learning process more susceptible to overfitting.

**Our Dirty-label Poisoning Attack.** Given the label of the target group $t$, the shadow dataset $\mathcal{D}_{\text{shadow}}$, and a poisoning budget $b_{\text{poison}}$, the poisoning dataset is constructed by the following steps: (1) select all samples $(x, t) \in \mathcal{D}_{\text{shadow}}$ with label $t$; (2) for each sample $(x, t)$, randomly change the label to another class $i \neq t$; (3) preserve at most $b_{\text{poison}}$ samples $(x, i)$ to compose $\mathcal{D}_{\text{poison}}$.

---

[3]We also evaluate our poisoning attacks with a stronger membership inference attack [9] on the CIFAR-10 dataset. The experimental results show that our poisoning attacks can also help improve the membership inference accuracy by a stronger membership inference attacker. We recommend interested readers to find more results in the appendix. In the main body of the paper, our experiments only assume the weak attack used by [40], as it has lower computational costs and is more practical.

Intuitively, only learning general concepts/features is insufficient to discriminate the clean samples of class $t$ from poisoning samples, since, in fact, they share similar features. To minimize the training loss, the victim model has to learn more specific features of each sample. As conceptually illustrated by Figure 1, the generalization performance on class $t$ tends to degrade. Figure 2 gives an example to show how our poisoning attack improves the membership exposure of the *airplane* category for an InceptionV3-based CIFAR-10 classifier. In this example, we assume the numbers of members and non-members are both 10,000 and $b_{\text{poison}}$=1,000.

**Remark.** We do not pose extra assumptions to the training process in this case. Our dirty-label poisoning attack is generic to all supervised learning scenarios, including end-to-end learning and transfer learning. Similar to the concurrent work [43], we also use the classical *label flipping* strategy [7] to carry out the dirty-label attack.

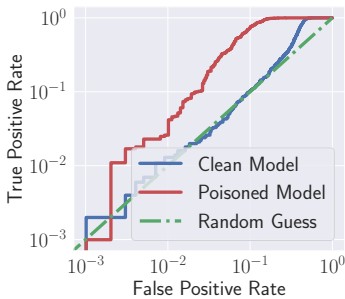

Figure 2: Membership inference against the *airplane* class of CIFAR-10 dataset. The poisoning budget is set as $b_{\text{poison}} = 10\% \times |\mathcal{D}_{\text{clean}}|$. The AUC score, i.e., the area under the ROC curve, is increased from 0.6917 to 0.9255, indicating that our dirty-label poisoning attack is effective.

## 3.2 Clean-label Poisoning Attack: A Stealthy Approach

In this part, we aim to improve our poisoning attack to amplify membership exposure while attaining stealthiness. Following the basic setup in [34], we consider the transfer learning setting. In transfer learning, we assume the victim model is composed of two parts: a pretrained feature extractor $g(\cdot)$ to extract high-level features from the input $x$ and a newly-trained classifier $c(\cdot)$ to predict labels.

The key to our clean-label poisoning is to exploit one vulnerability of deep feature extractors: a slight change in the input space may cause a significant change in the feature space. We propose the following attack strategy.

**Our Clean-label Poisoning Attack.** Given the label of the target class $t$, the shadow dataset $\mathcal{D}_{\text{shadow}}$, a poisoning budget $b_{\text{poison}}$, and the adopted feature extractor $g(\cdot)$, the poisoning dataset is constructed by the following steps: (1) from $\mathcal{D}_{\text{shadow}}$, select a *base sample* $(x_{\text{base}}, t)$ with label $t$ and a sample $(x, y)$ where $y \neq t$; (2) find a $x^*$ that is close to $x$ in the input space ($\|x^* - x\|_p \leq \epsilon$) and close to $x_{\text{base}}$ in the feature space ($g(x^*) \approx g(x_{\text{base}})$), and insert $(x^*, y)$ to $\mathcal{D}_{\text{poison}}$; (3) repeat step (1) and (2) until $|\mathcal{D}_{\text{poison}}| = b_{\text{poison}}$.

The intuition of our clean-label poisoning attack is straightforward: we actually mount a dirty-label poisoning attack in the feature space. That is, the poisoning sample $(x^*, y)$ is close to $(x_{\text{base}}, t)$, a clean sample from class $t$, in the feature space but labeled as $y$. We formalize step (2) as a constrained optimization problem:

$$x^* = \arg\min_{x'} \|g(x') - g(x_{\text{base}})\|_2, \text{s.t. } \|x' - x\|_\infty \leq \epsilon \text{ and } x' \in \mathcal{X} \tag{3}$$

where $\mathcal{X}$ refers to the input space. For instance, for a normalized RGB image input, a valid pixel value in one color channel should be within $[0, 1]$. For ease of optimization, we adopt $L_2$-norm to measure the distance in the feature space. Following the convention of adversarial attack work, we adopt $L_\infty$ to measure the distance in the input space. In our case, the constraint can be expressed as $x' \in [x_{\text{min}}, x_{\text{max}}]$, where $x_{\text{min}} = \max(0, x - \epsilon)$, $x_{\text{max}} = \min(1, x + \epsilon)$.

**Optimization.** We adopt the variable substitution method introduced by [11] to solve Equation 3. We introduce a new variable $w$ and let

$$\tanh(w) = \frac{2(x' - x_{\text{min}})}{x_{\text{max}} - x_{\text{min}}} - 1 \tag{4}$$

Based on the fact that $\tanh(w) \in [-1, 1]$, we can prove that the constraint $x' \in [x_{\text{min}}, x_{\text{max}}]$ holds in Equation 4. Plugging Equation 4 into Equation 3, we can obtain

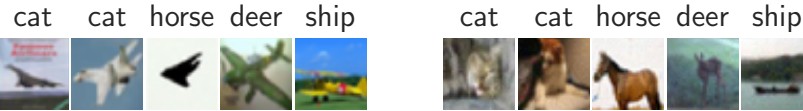

Figure 3: Examples of poisoning samples against the *airplane* category. **Left:** Dirty-label poisoning samples. **Right:** Clean-label poisoning samples ($\epsilon$=16/255). We provide the label of each sample on the top. For the clean-label poisoning samples, they look like natural ones.

$$w^* = \arg\min_{w} \|g\left(0.5 * (\tanh(w) + 1)(x_{\max} - x_{\min}) + x_{\min}\right) - g(x_{\text{base}})\|_2,$$
$$\text{and } x^* = 0.5(\tanh(w^*) + 1)(x_{\max} - x_{\min}) + x_{\min} \tag{5}$$

which poses an unconstrained optimization problem that can be simply solved by mainstream gradient optimization methods. In our implementation, we first initialize variable $w$ with $x$ by Equation 4 and iterate the optimization process 1,000 times, using an Adam optimizer with the learning rate 0.01.

Figure 3 illustrates some examples generated by our proposed two poisoning attacks. We can see that the poisoning samples produced by our clean-label poisoning method look natural to human eyes, which are hard to detect by manual moderation.

## 4 Evaluation

### 4.1 Experimental Setup

**Datasets.** In our experiment, we split each dataset into three portions: the clean training dataset $\mathcal{D}_{\text{clean}}$ containing the members, the testing dataset $\mathcal{D}_{\text{test}}$ containing the non-members, and the shadow dataset $\mathcal{D}_{\text{shadow}}$ for generating poisoning samples. We follow the same setup with [38], where $|\mathcal{D}_{\text{clean}}| = |\mathcal{D}_{\text{test}}| = |\mathcal{D}_{\text{shadow}}|$, and each of them does not overlap with others. Additionally, we set the three datasets to be balanced for the simplicity of evaluation among each class. We adopt five datasets for our experiments, including (1) **MNIST** [1] that contains 60,000 handwritten digits from 0 to 9. (2) **CIFAR-10** [2] that contains 60,000 images from 10 classes. (3) **STL-10** [3] that contains 13,000 labeled images from 10 classes. (4) **CelebA** [25] that contains 202,599 face images annotated by 40 attributes. In our experiment, we train a two-class classifier to predict the *Charming* attribute; (5) **PatchCamelyon** [46] that contains 327,680 images to predict the presence of metastatic tissue. For MNIST, CIFAR-10, CelebA, and PatchCamelyon dataset, we set $|\mathcal{D}_{\text{clean}}|$=10,000, while for STL-10, we set $|\mathcal{D}_{\text{clean}}|$=4,000. Each sample is transformed into a $96 \times 96 \times 3$ RGB image.

**Models.** We use five pretrained models provided by Tensorflow(v2.5.2): Xception, ResNet18, MobileNetv2, InceptionV3, and VGG16. For each model, we remove the fully connected (FC) layers to build up the feature extractor $g(\cdot)$. Then, we add two FC layers on top of the feature extractor to form $c(\cdot)$: one layer with 128 hidden units using the Tanh activation function, followed by an output layer using the Softmax activation function. We fix the feature extractor and train the newly added FC layers with the Adam optimizer, with the learning rate of $10^{-3}$ and batch size of 100.

**Poisoning Setup.** For each learning task, we mount the poisoning attack against each class. We set the poisoning budget $b_{\text{poison}}$ as $\frac{|\mathcal{D}_{\text{clean}}|}{\#\text{classes}}$. To make our poisoning attack stealthy to the victim, we assume $\mathcal{D}_{\text{poison}}$ is evenly distributed among all classes with each having approximately $\frac{b_{\text{poison}}}{\#\text{classes}}$ poisoned data samples. For instance, for a CIFAR-10 classifier, we set $b_{\text{poison}} = 1,000$, where only 100 samples for each class are in $\mathcal{D}_{\text{poison}}$. Note that this assumption means that there are $\frac{b_{\text{poison}}}{\#\text{classes}}$ clean samples in $\mathcal{D}_{\text{poison}}$ for the target class $t$. For the clean-label poisoning dataset, we set the perturbation constraint $\epsilon = 16/255$.

**Equipment.** Our experiments were conducted on a deep learning server, which is equipped with an Intel(R) Xeon(R) Gold 6226R CPU @ 2.90GHz, 128GB RAM, and four NVIDIA GeForce RTX 3090 GPUs with 24GB of memory.

| | Without Poisoning | | | Dirty-Label Poisoning | | | Clean-Label Poisoning | | |
|---|---|---|---|---|---|---|---|---|---|
| | MI AUC | TPR@FPR=1% | Test Acc. | MI AUC | TPR@FPR=1% | Test Acc. | MI AUC | TPR@FPR=1% | Test Acc. |
| **MNIST** | | | | | | | | | |
| Xception | .538±.022 | 1.22±0.56% | .939 | .697±.020 | 4.12±1.49% | .918±.007 | .627±.032 | 1.50±0.72% | .924±.005 |
| InceptionV3 | .546±.029 | 1.01±0.39% | .928 | .791±.026 | 3.52±1.21% | .902±.002 | .674±.053 | 1.34±0.61% | .910±.005 |
| VGG16 | .536±.021 | 1.24±0.55% | .954 | .740±.029 | 3.77±1.34% | .934±.004 | .604±.033 | 0.94±0.35% | .943±.002 |
| ResNet50 | .525±.021 | 1.22±0.62% | .967 | .721±.034 | 3.68±1.56% | .946±.009 | .583±.022 | 1.38±0.33% | .963±.002 |
| MobileNetV2 | .529±.019 | 1.04±0.49% | .960 | .759±.043 | 3.91±1.06% | .937±.004 | .626±.039 | 1.60±0.58% | .949±.004 |
| **CIFAR-10** | | | | | | | | | |
| Xception | .642±.057 | 1.21±0.55% | .768 | .893±.018 | 3.08±1.21% | .735±.004 | .868±.032 | 2.46±0.56% | .738±.004 |
| InceptionV3 | .733±.057 | 1.35±0.46% | .677 | .935±.015 | 7.28±2.83% | .648±.005 | .827±.046 | 1.44±0.59% | .663±.001 |
| VGG16 | .619±.049 | 1.12±0.40% | .815 | .899±.011 | 4.60±1.26% | .779±.004 | .869±.019 | 2.69±0.76% | .783±.003 |
| ResNet50 | .597±.035 | 1.06±0.29% | .848 | .930±.015 | 7.38±2.77% | .832±.003 | .861±.042 | 1.73±0.77% | .838±.002 |
| MobileNetV2 | .602±.050 | 1.20±0.66% | .842 | .916±.011 | 3.15±1.44% | .813±.002 | .836±.037 | 2.03±0.63% | .820±.002 |
| **STL-10** | | | | | | | | | |
| Xception | .578±.043 | 1.50±0.91% | .857 | .906±.021 | 6.83±2.92% | .836±.004 | .876±.027 | 3.90±1.63% | .838±.004 |
| InceptionV3 | .696±.069 | 1.57±0.78% | .758 | .940±.020 | 12.80±8.09% | .735±.006 | .809±.065 | 1.60±1.44% | .750±.002 |
| VGG16 | .596±.040 | 1.00±0.66% | .875 | .895±.017 | 8.07±6.36% | .842±.004 | .858±.029 | 3.80±2.71% | .846±.005 |
| ResNet50 | .572±.031 | 1.65±0.79% | .897 | .931±.015 | 11.28±10.28% | .875±.005 | .860±.041 | 2.83±1.44% | .883±.003 |
| MobileNetV2 | .558±.033 | 1.12±0.92% | .935 | .900±.019 | 4.35±3.35% | .906±.004 | .822±.042 | 2.08±1.27% | .916±.004 |
| **CelebA** | | | | | | | | | |
| Xception | .644±.011 | 1.01±0.01% | .724 | .752±.001 | 2.90±1.10% | .693±.002 | .710±.027 | 2.61±0.69% | .689±.005 |
| InceptionV3 | .748±.019 | 1.10±0.28% | .687 | .849±.008 | 2.75±0.01% | .655±.000 | .759±.056 | 1.58±0.30% | .661±.001 |
| VGG16 | .711±.019 | 1.26±0.06% | .724 | .747±.005 | 2.29±0.35% | .684±.003 | .713±.005 | 1.37±0.17% | .680±.003 |
| ResNet50 | .571±.000 | 0.89±0.09% | .744 | .680±.007 | 1.92±0.04% | .698±.007 | .616±.049 | 1.13±0.13% | .712±.000 |
| MobileNetV2 | .680±.009 | 1.21±0.11% | .750 | .823±.002 | 2.22±0.04% | .704±.002 | .742±.031 | 1.44±0.16% | .696±.006 |
| **PatchCamelyon** | | | | | | | | | |
| Xception | .564±.003 | 1.18±0.04% | .847 | .678±.020 | 2.12±0.22% | .797±.002 | .644±.007 | 1.65±0.13% | .816±.001 |
| InceptionV3 | .617±.008 | 0.94±0.10% | .832 | .739±.029 | 2.38±0.02% | .774±.008 | .627±.038 | 1.13±0.05% | .800±.001 |
| VGG16 | .538±.003 | 1.15±0.05% | .862 | .623±.016 | 1.69±0.13% | .842±.004 | .593±.004 | 1.25±0.11% | .838±.001 |
| ResNet50 | .543±.005 | 1.35±0.17% | .891 | .701±.038 | 2.25±0.29% | .820±.015 | .611±.017 | 1.47±0.11% | .862±.002 |
| MobileNetV2 | .565±.004 | 1.04±0.18% | .890 | .728±.029 | 1.79±0.07% | .819±.000 | .674±.001 | 1.17±0.05% | .842±.007 |

Table 1: Membership inference (MI) results for the target class $t$ and test-time accuracy on $\mathcal{D}_{\text{test}}$, for models not poisoned, poisoned by dirty-label attacks, and poisoned by clean-label attacks, respectively. We run the attack and evaluation over each class, and we report the average value with standard deviation for each metric.

## 4.2 Results

Table 1 reports the evaluation results on our proposed two poisoning attacks. In general, we can observe that our attacks obviously increase the AUC score of the membership inference against the target class $t$, with slight testing accuracy decay. For instance, for the clean-label poisoning attack against the STL-10 Xception-based classifier, the membership inference AUC is increased from 0.578 to 0.876 on average, while the testing accuracy is decreased from 0.857 to 0.838 on average. The results show that the poisoned samples resemble the clean samples and the performance of the poisoned models degrades subtly on the testing dataset. Consequently, there are chances that the clean-label poisoning attack evades the manual inspection, when the classification performance for each class does not get carefully examined.

Another observation is that the dirty-label poisoning attack causes more significant membership exposure than the clean-label poisoning attack. A potential explanation of this phenomenon is that our clean-label poisoning attack can be considered as an "approximate" version of the dirty-label poisoning attack in the feature space.

# 5 Ablation Study

## 5.1 Study 1: Impact of $\mathcal{D}_{\text{shadow}}$

In our prior study, we assume the attacker can obtain a shadow dataset $\mathcal{D}_{\text{shadow}}$ with the same distribution of the victim's clean training dataset $\mathcal{D}_{\text{clean}}$. However, the model developer may hide the training dataset information to fortify privacy and intellectual property (IP) protection. In this ablation section, we examine how the $\mathcal{D}_{\text{shadow}}$ affects the attack performance.

In our experiment, we assume the attacker aims to poison the *airplane* and *cat* categories against the CIFAR-10 classifiers respectively. For each target class $t$, we select 1,000 samples from $t$ in

| | $t=0$ (airplane) | | | $t=3$ (cat) | | |
|---|---|---|---|---|---|---|
| | Δ MI AUC | Δ TPR@TPR=1% | Δ Test Acc. | Δ MI AUC | Δ TPR@TPR=1% | Δ Test Acc. |
| Xception | .058 | -0.80% | .000 | .065 | 0.00% | -.006 |
| InceptionV3 | .066 | 0.00% | -.013 | .045 | 0.60% | -.007 |
| VGG16 | .057 | 0.30% | -.016 | -.009 | -0.10% | -.003 |
| ResNet50 | .019 | -0.50% | -.009 | -.017 | -0.10% | -.007 |
| MobileNetV2 | .019 | -0.20% | .000 | .000 | -0.10% | -.001 |

Table 2: Dirty-poisoning attack results when $\mathcal{D}_{\text{clean}}$ comes from CIFAR-10 and $\mathcal{D}_{\text{shadow}}$ contains 1,000 instances from the target class $t$ of STL-10. We report the membership exposure difference and testing accuracy difference between the poisoned model and the clean model.

| | Without Poisoning | | | Dirty-Label Poisoning | | | Clean-Label Poisoning | | |
|---|---|---|---|---|---|---|---|---|---|
| | MI AUC | TPR@FPR=1% | Test Acc. | MI AUC | TPR@FPR=1% | Test Acc. | MI AUC | TPR@FPR=1% | Test Acc. |
| InceptionV3 | .619±.053 | 1.32±0.57% | .862 | .956±.008 | 15.17±6.42% | .845±.002 | .618±.054 | 1.20±0.43% | .863±.001 |
| MobileNetV2 | .553±.026 | 1.27±0.70% | .906 | .820±.021 | 4.76±1.17% | .896±.002 | .556±.024 | 1.11±0.41% | .909±.001 |
| Xception | .573±.030 | 1.08±0.51% | .898 | .923±.007 | 9.02±2.71% | .878±.002 | .581±.030 | 1.26±0.37% | .898±.001 |
| VGG16 | .622±.052 | 1.04±0.38% | .875 | .904±.010 | 13.98±5.07% | .858±.002 | .622±.060 | 1.58±0.92% | .874±.002 |
| ResNet50 | .584±.030 | 1.26±0.43% | .903 | .942±.009 | 13.37±4.40% | .884±.002 | .592±.032 | 1.30±0.66% | .901±.001 |

Table 3: Dirty-label and clean-label attack results on fine-tuned CIFAR-10 classifiers ($b_{\text{poison}} = 1,000$). We run the attack and evaluation over each class, and we report the average value with standard deviation for each metric.

STL-10. In this ablation study, we start with the dirty-label poisoning attack to estimate the upper bound performance of our clean-label poisoning attack. Table 2 reports the attack performance. Our results show that the attack performance declines when the attacker's shadow data has a different distribution from the actual clean dataset.

## 5.2 Study 2: Impact of $b_{\text{poison}}$

We then study how the size of poisoning samples affects the attack performance. Figure 4 depicts the membership exposure under dirty-label poisoning attacks with different poisoning budget $b_{\text{poison}}$. Unsurprisingly, as we gradually decrease the poisoning budget, the membership inference AUC score decreases as well. Yet, it is still possible to achieve a significant membership exposure increase even with a small poison budget. Take ResNet50 for instance, when $b_{\text{poison}}$ is 100, i.e., the amount of poisoning samples is only 1% of that for the clean samples, we can improve the membership inference AUC score from 0.5971 to 0.7076 on average.

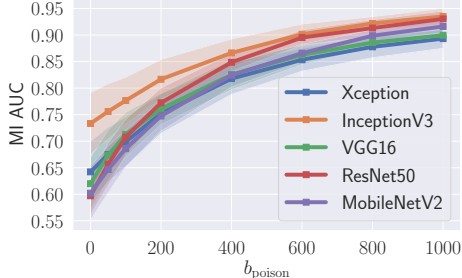

Figure 4: Membership exposure under different $b_{\text{poison}}$. We plot the average MI AUC on the CIFAR-10 dataset.

## 5.3 Impact of Fine-tuning

In the transfer learning scenario, there are also chances that model developers fine-tune parameters of feature extractors [14, 36, 47]. The fine-tuning operation is supposed to make the transfer learning model better fit the new dataset. Inevitably, it may affect how the feature extractor extracts latent features from the inputs. As a result, our poisoning attack may get affected. In this part, we explore how the fine-tuning process impacts our attack performance.

In our experiment, we train CIFAR-10 classifiers with fine-tuning on the poisoned dataset, with the learning rate set as 1e-5. We report the attack performance in Table 3. It can be seen that, for our dirty-label poisoning attack, we achieve more significant membership exposure amplifications. While for our clean-label poisoning attack, the membership exposure changes are nearly negligible.

To understand this phenomenon, we investigate how the feature extractor behaves in different cases. We visualize the latent features of clean samples and poisoning samples by t-SNE technique [45] in Figure 5. We can observe that, for the dirty-label poisoning attack in the fine-tuning setting, the poisoning samples distribute closely to the targeted clean samples in the feature space (as shown

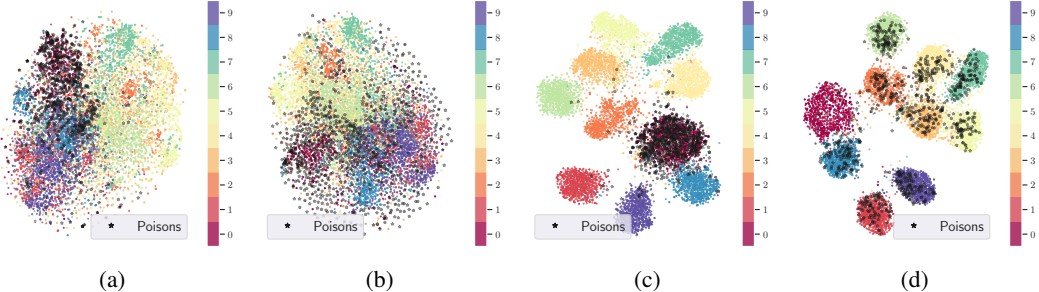

| (a) | (b) | (c) | (d) |

Figure 5: Visualization of latent features extracted by different poisoned CIFAR-10 classifiers from the InceptionV3 feature extractor. The targeted class is 0 (*airplane*). (a) Dirty-label attack against a fixed feature extractor. (b) Clean-label attack against a fixed feature extractor. (c) Dirty-label attack against a fine-tuned model. (d) Clean-label attack against a fine-tuned model. The colored points are the clean training samples, while the dark star marks are the poisoning sample.

in Figure 5c. Besides, for the clean-label poisoning attack in the fine-tuning setting, the poisoning samples distribute among each class in the feature space (as shown in Figure 5d). It indicates that the more poisoning samples in the target class, the more significant membership exposure results we can get. By comparing Figure 5a and Figure 5b, we can also partially illustrate that dirty-label poisoning tends to achieve better attack performance than clean-label poisoning. This trend is actually consistent with the intuition of our attack. In fact, this is the case called "adversarial training" by some adversarial example defense literature [35, 44, 49]. During adversarial training, model parameters are gradually tuned to eliminate the effects of input perturbations.

## 6 Discussion on Potential Countermeasures

In general, there are two potential ways to defend our poisoning attacks: One is to detect and filter out poisoning data, while the other is to reduce membership exposure. We will investigate the former direction in the future, and in this part, we mainly explore defenses by limiting membership information leakage, including:

- **Regularization.** Overfitting is considered to be one of the major culprits of membership exposure [38, 51, 33]. Therefore, the regularization technique may be feasible to defend against our attacks. In our experiment, we introduce an $L_2$-norm regularizer with a penalty of 0.05 during the model training process.

- **Early Stopping.** Early stopping is another common practice to prevent overfitting [21]. During the training process, for each epoch, we randomly sample out 10% of the training data as validation data and use 90% of other data as training data. We monitor the validation loss, and if it does not decrease in three epochs, we stop the training process.

- **DP-SGD.** Differential privacy (DP) provides a rigorous guarantee to limit privacy leakage [12, 13]. Recently, privacy-preserving machine learning algorithms under the differential privacy have been proposed [6, 18, 41], among which differentially private stochastic descent (DP-SGD) [4] receives the most attention [5, 20, 28]. In our experiment, we utilize the DP-SGD optimizer provided by the TensorFlow Privacy package[4] to implement differential privacy training. The hyperparameters used in our implementation are summarized in Table 4. As reported by the analysis tool provided by TensorFlow Privacy, we achieve $(3.25, 10^{-6})$-differential privacy on the clean model ($|\mathcal{D}_{\text{train}}|=|\mathcal{D}_{\text{clean}}|$=10,000), while we achieve $(3.10, 10^{-6})$-differential privacy on poisoned models ($b_{\text{poison}} = 1,000$, $|\mathcal{D}_{\text{train}}|=|\mathcal{D}_{\text{clean}} \cup \mathcal{D}_{\text{poison}}|$=11,000).

We evaluate the three potential countermeasures on the CIFAR-10 dataset and summarize the results in Table 5. By comparing the membership exposure between unprotected (Table 1) and protected models (Table 5), the three defenses can help weaken the membership exposure amplification effect by our attacks. Besides, in our experiment, we observe that the regularization technique has the best

---

[4]https://github.com/tensorflow/privacy

| Hyperparameter | Value |
|---|---|
| LEARNING RATE | 0.001 for InceptionV3; 0.01 for others |
| NOISE MULTIPLIER | 1.0 |
| MAX $L_2$-NORM OF GRADIENTS | 1.0 |
| BATCH SIZE | 100 |
| MICRO BATCH SIZE | 100 |
| EPOCHS | 20 |

Table 4: Hyperparameters used by the differential private training algorithms.

| | Without Poisoning | | | Dirty-Label Poisoning | | | Clean-Label Poisoning | | |
|---|---|---|---|---|---|---|---|---|---|
| | MI AUC | TPR@FPR=1% | Test Acc. | MI AUC | TPR@FPR=1% | Test Acc. | MI AUC | TPR@FPR=1% | Test Acc. |
| **Early Stopping** | | | | | | | | | |
| InceptionV3 | .598±.045 | 0.95±0.23% | .622 | .614±.034 | 1.17±0.37% | .674±.001 | .621±.031 | 1.09±0.30% | .676±.003 |
| MobileNetV2 | .561±.036 | 1.40±0.48% | .758 | .560±.026 | 1.33±0.58% | .840±.000 | .560±.026 | 1.40±0.54% | .840±.000 |
| Xception | .567±.043 | 1.00±0.44% | .699 | .576±.031 | 1.19±0.37% | .769±.000 | .576±.031 | 1.18±0.36% | .769±.000 |
| VGG16 | .560±.036 | 1.12±0.44% | .728 | .570±.030 | 1.22±0.47% | .798±.003 | .571±.029 | 1.29±0.38% | .798±.000 |
| ResNet50 | .561±.037 | 0.88±0.26% | .771 | .566±.026 | 1.07±0.34% | .849±.001 | .569±.025 | 1.08±0.31% | .851±.000 |
| **Regularization** | | | | | | | | | |
| InceptionV3 | .524±.011 | 1.01±0.42% | .642 | .522±.011 | 1.09±0.56% | .639±.006 | .525±.008 | 1.01±0.41% | .643±.005 |
| MobileNetV2 | .517±.010 | 1.32±0.56% | .797 | .522±.008 | 1.12±0.32% | .801±.009 | .525±.008 | 1.11±0.33% | .798±.007 |
| Xception | .513±.011 | 1.21±0.43% | .722 | .514±.013 | 1.31±0.67% | .702±.008 | .514±.014 | 1.29±0.83% | .702±.006 |
| VGG16 | .524±.012 | 1.34±0.50% | .792 | .525±.009 | 1.48±0.60% | .803±.003 | .538±.012 | 1.57±0.55% | .797±.002 |
| ResNet50 | .519±.008 | 1.22±0.52% | .828 | .521±.010 | 1.18±0.21% | .806±.009 | .527±.010 | 1.41±0.48% | .801±.008 |
| **DP-SGD** | | | | | | | | | |
| InceptionV3 | .527±.015 | 1.16±0.52% | .616 | .531±.016 | 1.32±0.47% | .613±.001 | .529±.015 | 1.26±0.51% | .617±.001 |
| MobileNetV2 | .525±.012 | 0.14±0.42% | .689 | .539±.007 | 1.52±0.73% | .726±.003 | .538±.009 | 1.62±0.74% | .728±.002 |
| Xception | .526±.012 | 0.81±0.79% | .653 | .540±.012 | 1.12±0.35% | .683±.001 | .541±.009 | 1.07±0.36% | .682±.002 |
| VGG16 | .514±.011 | 0.92±0.83% | .665 | .525±.008 | 1.24±0.44% | .691±.003 | .524±.009 | 1.45±0.59% | .693±.003 |
| ResNet50 | .518±.010 | 0.56±1.19% | .705 | .533±.009 | 1.04±0.31% | .735±.001 | .531±.008 | 1.13±0.57% | .736±.002 |

Table 5: Effectiveness of our poisoning attacks when student models are trained with early stopping, regularization, and DP-SGD countermeasures, respectively. The results come from CIFAR-10 with $b_{\text{poison}} = 1,000$. We run the attack and evaluation over each class, and we report the average value with standard deviation for each metric.

model utility-privacy trade-off. The regularization technique achieves a relatively high model testing accuracy and keeps the AUC scores at lower levels.

# 7 Conclusion

In this paper, we demonstrate the feasibility of exploiting data poisoning to amplify the membership exposure of the training dataset. We present attacks that significantly increase the precision of the membership inference attack on the targeted class, with limited negative influences on the model's test-time performance. We also conduct extensive evaluations to study how different factors may affect attack performance.

Our findings uncover a new challenge to modern machine learning ecosystems. Data from the open world may not only affect the model performance but also raise real threats to private training data. Even worse, we show it is possible to mount clean-label attacks to evade human moderation. Our research is thus a call to action. One primary limitation of our clean-label attacks is that we assume the attacker has knowledge of the feature extractor. How to create clean-label poisons in a black-box manner remains our future work.

## Acknowledgments and Disclosure of Funding

We thank the anonymous reviewers for their constructive comments. This work is supported by the National Key Research and Development Program of China (2020AAA0107702), National Natural Science Foundation of China (U21B2018, 62161160337, 62132011), Shaanxi Province Key Industry Innovation Program (2021ZDLGY01-02), the Research Grants Council of Hong Kong under Grants N_CityU139/21, R6021-20F, R1012-21, C2004-21G, and the Helmholtz Association within the project "Trustworthy Federated Data Analytics" (TFDA) (funding number ZT-I-OO1 4). Chao Shen, Cong Wang, and Yang Zhang are the corresponding authors.

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
