# A    Membership Exposure Evaluated by a Stronger Attack

In this section, we evaluate the membership exposure effect with a stronger membership inference attack proposed by [1]. Our experiment is conducted on the CIFAR-10 dataset. For each encoder, we have trained 128 shadow models, with 50% samples randomly selected from $\mathcal{D}_{\text{clean}} \bigcup \mathcal{D}_{\text{test}}$ (i.e., all the members and non-members in our setup). Table 6 reports the evaluation results. The results show that our attacks have increased the membership exposure risks.

| | Without Poisoning | | | Dirty-Label Poisoning | | | Clean-Label Poisoning | | |
|---|---|---|---|---|---|---|---|---|---|
| | MI AUC | TPR@FPR=1% | Test Acc. | MI AUC | TPR@FPR=1% | Test Acc. | MI AUC | TPR@FPR=1% | Test Acc. |
| Xception | .614±.047 | 0.38±0.20% | .768 | .675±.057 | 0.41±0.25% | .735±.004 | .677±.055 | 0.34±0.20% | .738±.004 |
| InceptionV3 | .712±.055 | 0.71±0.52% | .677 | .799±.053 | 0.94±0.57% | .648±.005 | .756±.054 | 0.93±0.62% | .663±.001 |
| VGG16 | .610±.041 | 0.71±0.66% | .815 | .707±.043 | 0.64±0.39% | .779±.004 | .710±.043 | 0.70±0.52% | .783±.003 |
| ResNet50 | .583±.021 | 0.56±0.35% | .848 | .694±.035 | 0.35±0.17% | .832±.003 | .679±.039 | 0.35±0.19% | .838±.002 |
| MobileNetV2 | .592±.039 | 0.34±0.20% | .842 | .674±.054 | 0.23±0.15% | .813±.002 | .659±.054 | 0.24±0.16% | .820±.002 |

Table 6: The effectiveness of our poisoning attacks evaluated by a stronger membership inference attack [1]. The results come from CIFAR-10 with $b_{\text{poison}} = 1,000$. We run the attack and evaluation over each class, and we report the average value with standard deviation for each metric.

# B    Towards Understanding Membership Exposure in the Experiments

To help understand the membership exposure phenomenon exhibited in the main text, we propose the following heuristic: for a sample $(x, y)$ from the training dataset $\mathcal{D}_{\text{train}}$, we define

$$h(x) = \frac{d_{\text{out}}(x) - d_{\text{in}}(x)}{\min\left(d_{\text{out}}(x), d_{\text{in}}(x)\right)} \tag{6}$$

where

$$d_{\text{in}}(x) = \min_{(x',y')\in\mathcal{D}_{\text{train}}\setminus\{x\},y'=y} \|x - x'\|_2 \tag{7}$$

$$d_{\text{out}}(x) = \min_{(x'',y'')\in\mathcal{D}_{\text{train}},y''\neq y} \|x - x''\|_2 \tag{8}$$

That is, $d_{\text{in}}(x)$ represents the distance from $x$ to its nearest neighbor within the same class, while $d_{\text{out}}(x)$ represents the distance from $x$ to its nearest neighbor from other classes. Intuitively, if a sample is closer to samples from other classes, it is more like an "outlier" and is prone to membership exposure [2, 1]. As a result, a lower $h$ indicates a higher membership exposure risk. In this section, we measure heuristic $h$ in the **feature space** established by the feature extractor $g(\cdot)$, which compensates for our findings in the main text.

## B.1    Case Study 1: Membership Exposure Across Different Feature Extractors

We plot $h$ across clean CIFAR-10 classifiers in Figure 6. It can be clearly seen that classifiers based on InceptionV3 have lower $h$ than other classifiers, which indicates that classifiers from InceptionV3 tend to have higher MI AUC. This is consistent with our results in Table 1.

## B.2    Case Study 2: Impact of $\mathcal{D}_{\text{shadow}}$

In Figure 7, We plot the cumulative distribution of $h$ over one clean and two poisoned CIFAR-10 InceptionV3 classifiers. The poisoned classifiers are poisoned by dirty-label attacks using STL-10 as the $\mathcal{D}_{\text{shadow}}$. It can be seen that the two CDF curves nearly overlap, indicating that dirty-label poisoning attacks based on STL-10 hardly affect CIFAR-10 classifiers. This is consistent with our results reported in Table 2.

## B.3    Case Study 3: Impact of Fine-tuning

We plot $h$ across CIFAR-10 classifiers fine-tuned from the InceptionV3 feature extractor in Figure 8. It can be clearly seen that there is a huge gap between the dirty-label poisoned models and the clean,

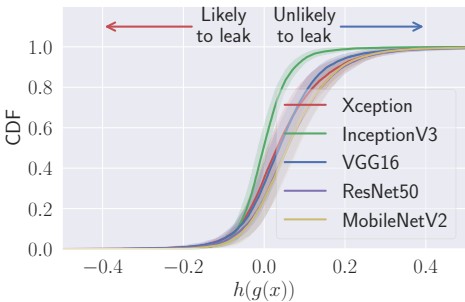

Figure 6: Cumulative distribution function (CDF) of the heuristic $h$ in the input space across clean CIFAR-10 classifiers. We compute $h$ for each target class and plot the average CDF.

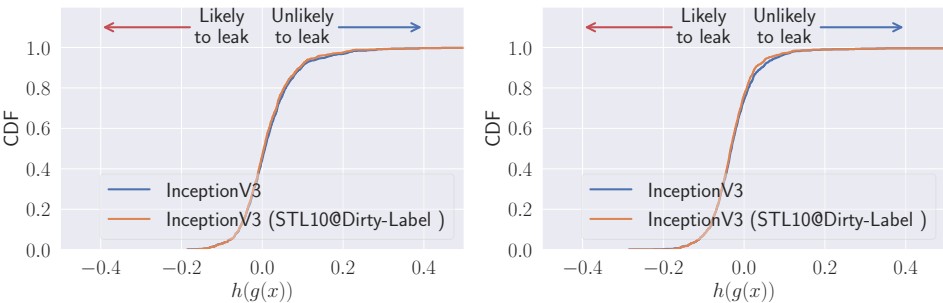

Figure 7: Dirty-label poisoning attack against CIFAR-10 classifiers when $\mathcal{D}_{\text{shadow}}$ is STL-10. **Left:** the *airplane* class. **Right:** the *cat* class.

fine-tuned models, indicating a significant MI AUC increase by the dirty-label poisoning attack. Meanwhile, the CDF curves between the clean models and the clean-label poisoned models nearly overlap, indicating that fine-tuning can help to resist clean-label poisoning attacks. This is consistent with our results in Table 3.

## C  Defense Through Differentially Private Training

In the main body of the paper, our results show that the differentially private training technique can effectively mitigate the membership exposure problem. To better understand how DP-SGD helps to defend our poisoning attacks, we also show the impact of DP-SGD on the learning process of clean models, models poisoned by dirty-label attacks, and models poisoned by clean-label attacks in Figure 9, Figure 10 and Figure 11, respectively.

**Defense Against Dirty-label Poisoning.** From Figure 10b, we can see that DP-SGD prevents models from learning from the poisoning samples: the training loss on poisoning samples increases and the training accuracy on poisoning samples remains at the random-guess level.

**Defense Against Clean-label Poisoning.** From Figure 11b, we can also observe a similar trend with dirty-label poisoning. DP-SGD prevents models from learning from the poisoning samples. However, different from dirty-label poisoning, training accuracy on the poisoning samples slightly improves in the training process. That is, a small portion of clean-label poisoning samples are correctly learned by the model. One potential explanation is that the dirty-label poisoning samples have a stronger impact on models to leak membership privacy. As a result, the DP-SGD puts more effort, i.e. adds more noise, to eliminate the effects of dirty-label poisoning samples, which would decrease the training accuracy. This is consistent with our finding that dirty-label poisoning attacks are likely to cause more significant membership exposure (see Section 4.2).

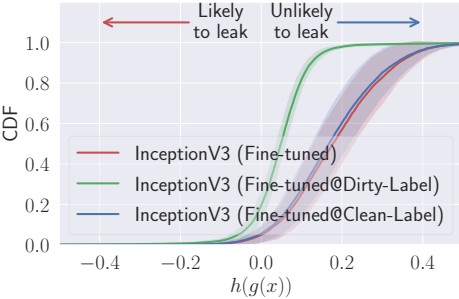

Figure 8: Impact of fine-tuning. We choose the InceptionV3-based classifier as the example.

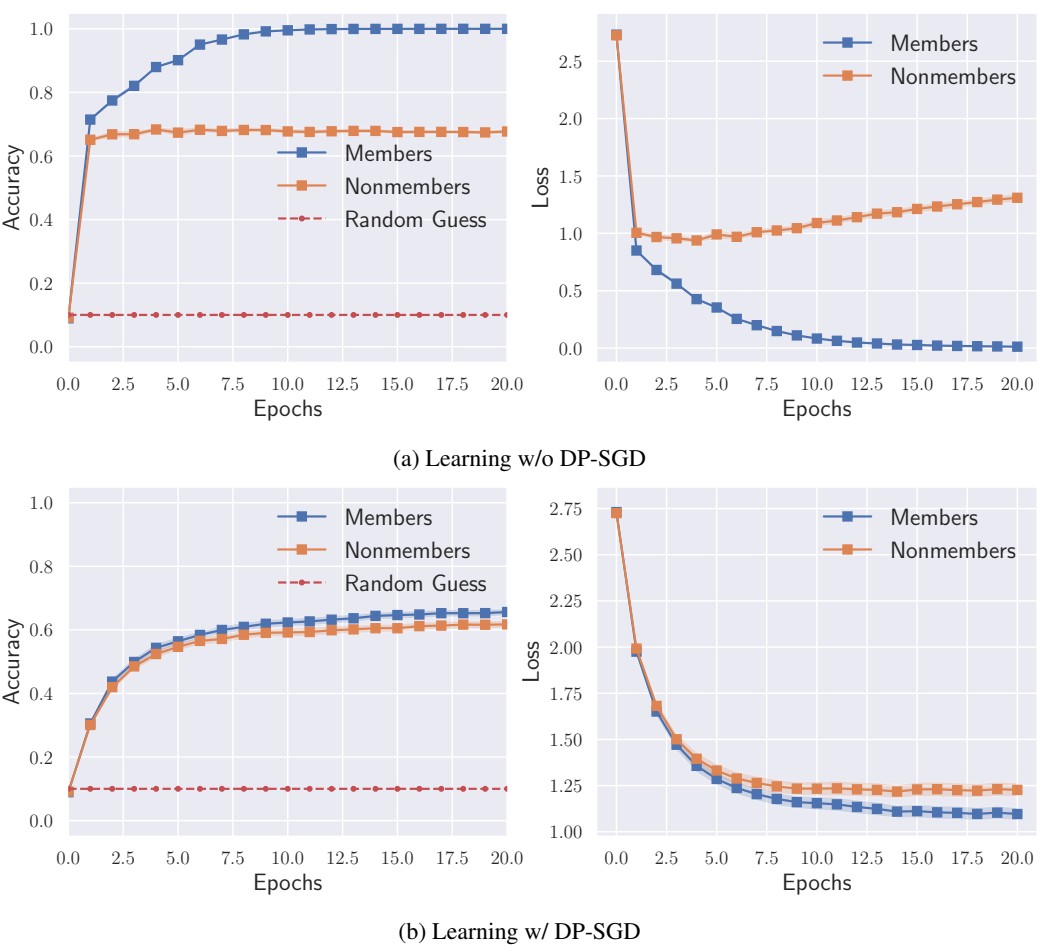

(a) Learning w/o DP-SGD

(b) Learning w/ DP-SGD

Figure 9: Learning process of clean models. The feature extractor is InceptionV3, and the dataset is CIFAR10. **Top:** Non-private training. **Bottom:** Training with DP-SGD.

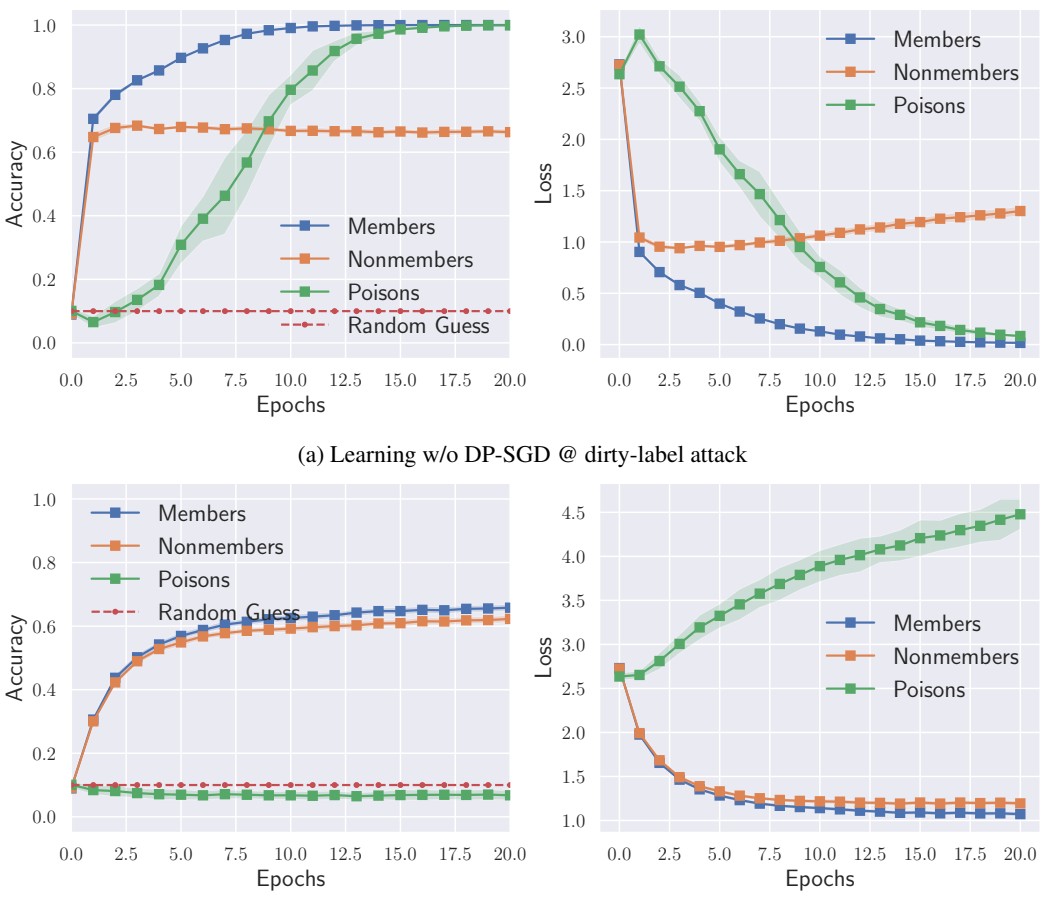

(a) Learning w/o DP-SGD @ dirty-label attack

(b) Learning w/ DP-SGD @ dirty-label attack

Figure 10: Learning process under the dirty-label poisoning attack. We run the attack against each class, where the feature extractor is InceptionV3, and the dataset is CIFAR10 with $b_{\text{poison}}$=1,000. **Top:** Non-private training. **Bottom:** Training with DP-SGD. We omit the normal examples in $\mathcal{D}_{\text{poison}}$.

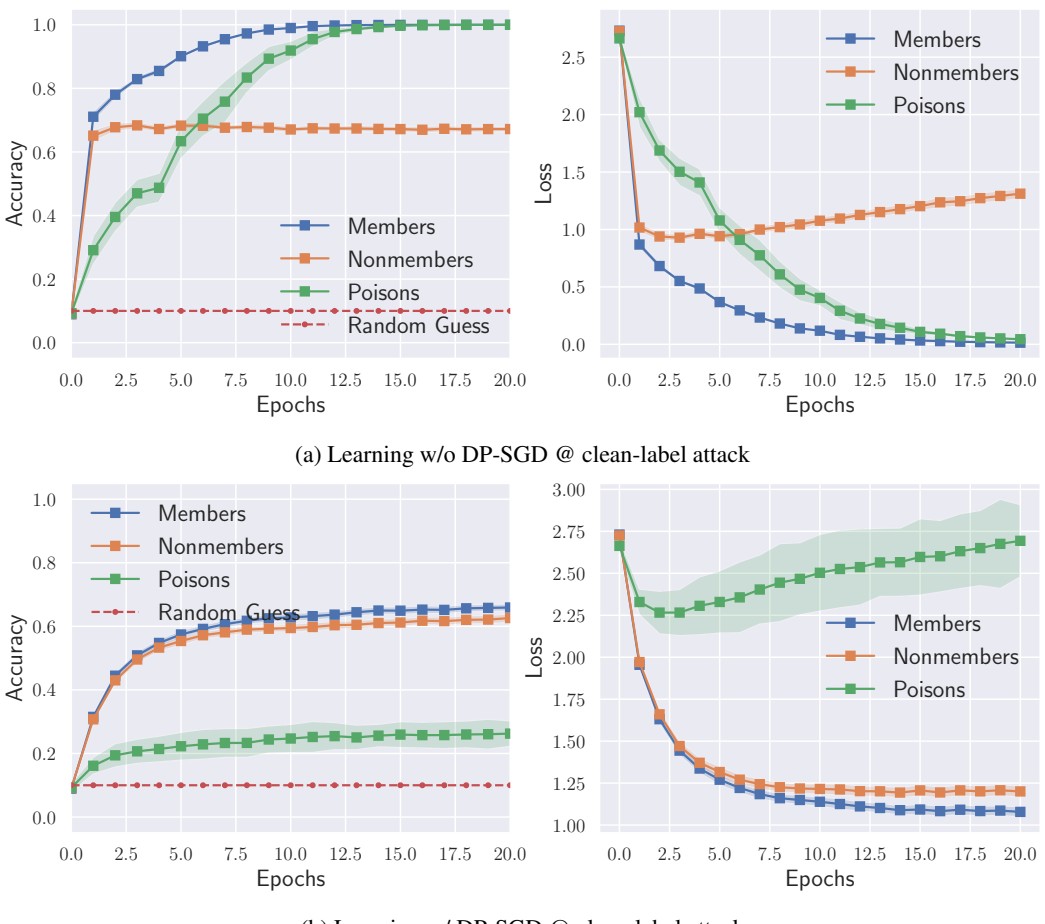

(a) Learning w/o DP-SGD @ clean-label attack

(b) Learning w/ DP-SGD @ clean-label attack

Figure 11: Learning process under the clean-label poisoning attack. We run the attack against each class, where the feature extractor is InceptionV3, and the dataset is CIFAR10 with $b_{\text{poison}}$=1,000. **Top:** Non-private training. **Bottom:** Training with DP-SGD. We omit the normal examples in $\mathcal{D}_{\text{poison}}$.

# D  Clean-Label Poisoning Examples

Here we exhibit some clean-label poisoning examples (Figure 12 for the MNIST dataset, Figure 13 for the CelebA dataset, Figure 14 for the CIFAR-10 dataset, Figure 15 for the PatchCamelyon dataset, and Figure 16 for the STL-10 dataset). They are generated from the InceptionV3 pretrained feature extractor. Note that all images are randomly selected rather than cherry-picked.

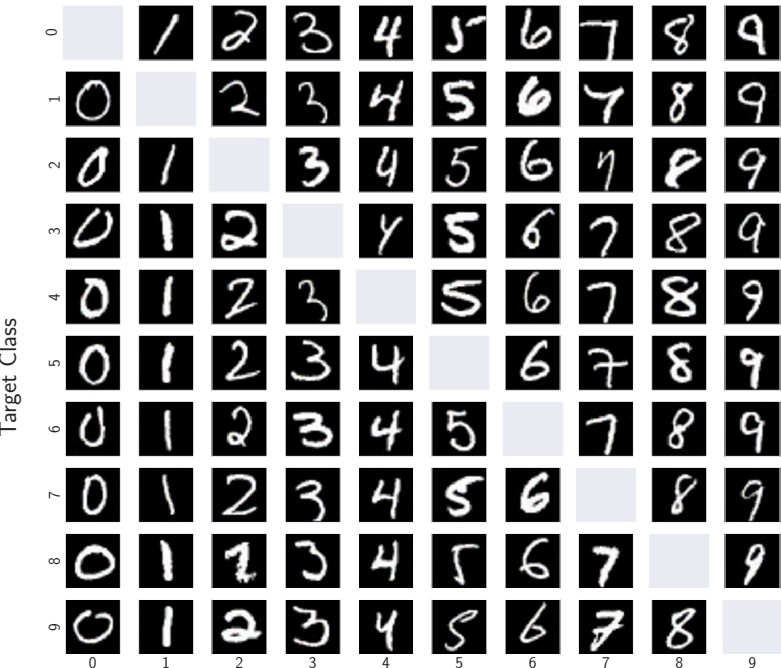

Figure 12: Clean-label poisoning examples for the MNIST dataset (generated from the InceptionV3 feature extractor, with $\epsilon = 16/255$). The labels along the x-axis refer to the labels of the poisoning samples, while the labels along the y-axis refer to the target class. We omit the normal examples in $\mathcal{D}_{\text{poison}}$, which are used to make the poisoning dataset look balanced.

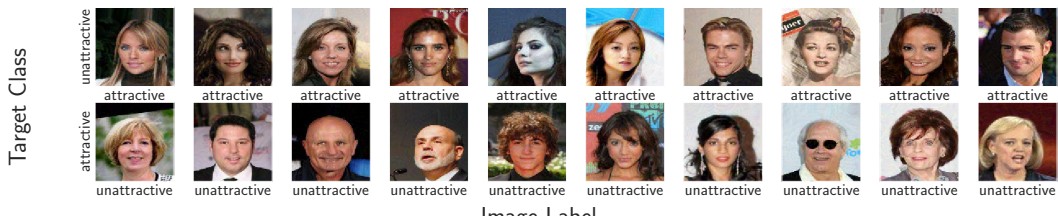

Figure 13: Clean-label poisoning examples for the CelebA dataset (generated from the InceptionV3 feature extractor, with $\epsilon = 16/255$). The labels along the x-axis refer to the labels of the poisoning samples, while the labels along the y-axis refer to the target class. We omit the normal examples in $\mathcal{D}_{\text{poison}}$, which are used to make the poisoning dataset look balanced.

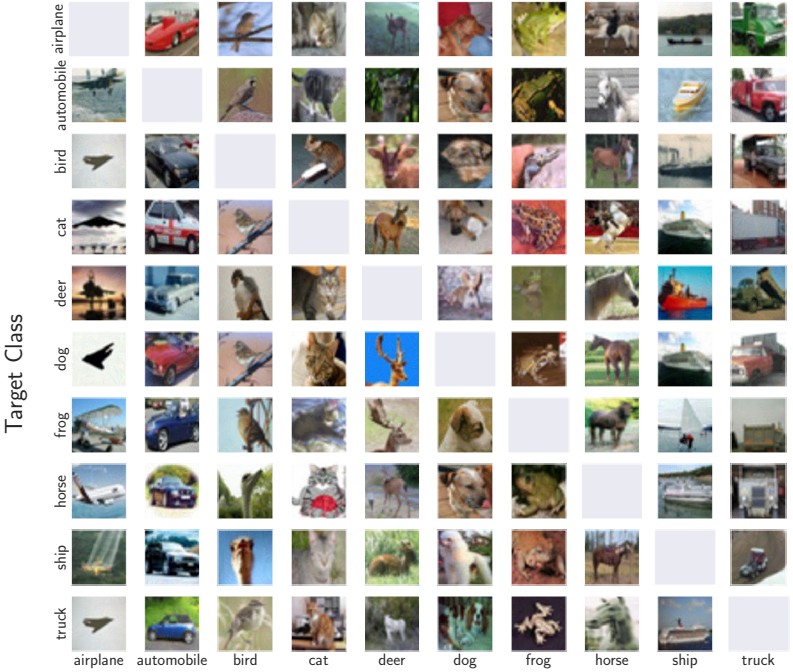

Figure 14: Clean-label poisoning examples for the CIFAR-10 dataset (generated from the InceptionV3 feature extractor, with $\epsilon = 16/255$). The labels along the x-axis refer to the labels of the poisoning samples, while the labels along the y-axis refer to the target class. We omit the normal examples in $\mathcal{D}_{\text{poison}}$, which are used to make the poisoning dataset look balanced.

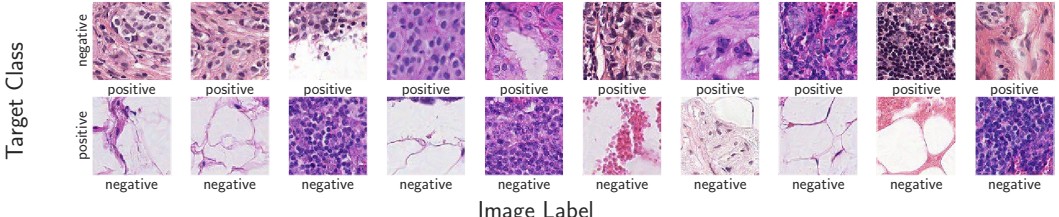

Figure 15: Clean-label poisoning examples for the PatchCamelyon dataset (generated from the InceptionV3 feature extractor, with $\epsilon = 16/255$). The labels along the x-axis refer to the labels of the poisoning samples, while the labels along the y-axis refer to the target class. We omit the normal examples in $\mathcal{D}_{\text{poison}}$, which are used to make the poisoning dataset look balanced.

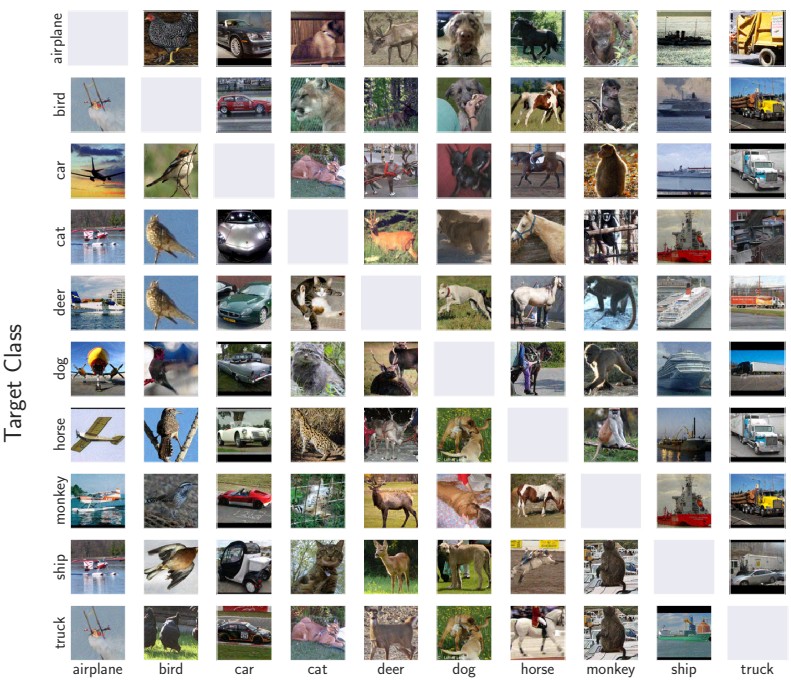

Figure 16: Clean-label poisoning examples for the STL-10 dataset (generated from the InceptionV3 feature extractor, with $\epsilon = 16/255$). The labels along the x-axis refer to the labels of the poisoning samples, while the labels along the y-axis refer to the target class. We omit the normal examples in $\mathcal{D}_{\text{poison}}$, which are used to make the poisoning dataset look balanced.