# OpenReview forum: "Amplifying Membership Exposure via Data Poisoning"
_NeurIPS.cc/2022/Conference — NeurIPS 2022 Accept_

### Official Review · Reviewer_gUo3 · 2022-07-06

**Rating:** 7
**Confidence:** 5
**Soundness:** 3 good
**Presentation:** 4 excellent
**Contribution:** 3 good

**Summary:**

This paper shows that poisoning a training set can amplify the privacy leakage (under membership inference attacks) of other training points.
The paper proposes a dirty-label and a clean-label attack and evaluates them in the transfer learning setting. Both attacks significantly increase the AUC of the membership inference attack.

**Questions:**

- In the dirty-label attack, do you use the same wrong class i for each poison, or is i sampled at random for each poison?
- Why such a big difference between the dirty-label and clean-label attacks?
- Can you show that the dirty-label attack actually works in an "end-to-end" training scenario?
- Can you show how well your MI attacks work at low false-positive rates?

**Limitations:**

N.A

**Strengths And Weaknesses:**

This paper studies a new and interesting threat model where the ability to poison a model is used to target privacy of other training points (rather than to evade the model).
The same problem is also considered in concurrent work [34], while the prior work of [18] considers weaker leakages of global dataset properties. (Calling [18] concurrent work is a bit of a stretch: even though it was only officially published this May, the paper has been on arxiv since Jan. 2021).

The paper's aim is to globally increase membership inference success rates, and it also evaluates attack success rates in an average-case sense, using the AUC.
It could be nice here to also consider worst-case membership inference metrics, as advocated in recent work by Carlini et al: https://arxiv.org/abs/2112.03570
As Figure 2 shows, the poisoning attack seems to not only increase the AUC, but to also drastically increase the attack's TPR at low FPRs, which is what really matters for a privacy attack.

My main criticism of the paper is that it only considers the transfer setting. Even the dirty-label attack which can in principle be applied in any setting is only evaluated in the transfer setting. It might be worth acknowledging this in the abstract and introduction.
In particular, conducting clean-label attacks in the transfer setting seems a bit trivial, as one can essentially simulate the dirty-label attack in feature space (the paper does acknowledge this).

So I was actually surprised to see such a big difference in the results for the dirty-label and clean-label attacks. It is not clear to me why this should be the case: with a perturbation of eps=16/255 it should be possible to near-perfectly match the features of any other example.
The visualization on Figure 5 suggests that the attacks actually work quite differently: it seems that the dirty-label attack always uses the same "dirty" label, whereas the clean-label attack does not. Can you confirm this? And if this is indeed the case, why? Is it just because you don't have enough data points of the same class in D_shadow?

Regarding the threat model, it might be worth commenting a bit on how the assumptions behind poisoning attacks and membership inference attacks interact. In particular, to run a membership inference attack the attacker must anyhow get access to some samples from the distribution (namely the samples on which the attacker will try to infer membership). So the assumption that D_shadow exists seems perfectly natural (and the study in 5.1 may not even be needed). In fact, in practice the adversary would likely know a D_shadow that intersects with D_clean (the adversary just doesn't know which points are in the intersection, which is precisely the membership inference problem).

---

> ### Author Response · Authors · 2022-08-02
> **Response to Reviewer gUo3**
>
> We thank the reviewer for their valuable comments (C). We hope that our responses (R) have fully addressed all of the reviewer’s concerns, and remain committed to clarifying any further questions that may arise during the discussion period.
>
> ***C1: My main criticism of the paper is that it only considers the transfer setting…***
>
> ***R1:***
> Thanks for your comments. We study the transfer learning setting as it has become a common practice to save time on collecting large datasets and training models from scratch, which is recommended by many popular frameworks, such as [Tensorflow](https://www.tensorflow.org/tutorials/images/transfer_learning) and [PyTorch](https://pytorch.org/tutorials/beginner/transfer_learning_tutorial.html). Despite its popularity, the transfer learning paradigm has been shown to have various security and privacy issues.
>
> One of the most well-known threats to transfer learning is the clean-label poisoning attack [1*][2*], wherein the attacker does not require the control of the labeling process and can craft samples with natural appearances to evade moderation. Our work is to complement existing findings for clean-label poisoning attacks, as we reveal a new threat by clean-label poisoning attacks: an attacker can exploit clean-label poisons to amplify membership exposures for normal training samples.
>
> Besides the transfer learning setting, our dirty-label attacks can actually work in the “end-to-end” training scenario. We follow your suggestions to evaluate this case. Please refer to R5 for more details.
>
> ***C2: Regarding the threat model, it might be worth commenting a bit on how the assumptions behind poisoning attacks and membership inference attacks interact…***
>
> ***R2:***
> Thanks for your comments. The ablation study in Section 5.1 shows that the distribution of $D_{shadow}$ would impact the poisoning performance, and our results show that if the distribution of $D_{shadow}$ is different from that of $D_{clean}$, the membership exposure increased by our attack is insignificant. The results can be intuitively interpreted as only when poisons get close to the normal samples can they have significant impacts.
>
> If $D_{shadow}$ intersects with $D_{clean}$, for these samples simultaneously in $D_{shadow}$ and $D_{clean}$, we would be likely to get higher TPR at low FPR (note: as shown by the targeted attack in the concurrent work [3*]). But for a more fair comparison, our evaluation setup still follows the assumption by prior MI attacks [4*][5*], where $D_{shadow}$ does not interset with $D_{clean}$. Thanks for your insightful comments, and we will investigate the attack performance when $D_{shadow}$ intersects with $D_{clean}$.
>
> ***C3: In the dirty-label attack, do you use the same wrong class i for each poison, or is i sampled at random for each poison?***
>
> ***R3:***
> In the dirty-label attack, we randomly distribute the wrong labels to each poison. Note that to make the poisoning dataset looks “balanced,” we keep the number of poisons in each wrong class $i$ nearly the same.
>
> Note: in Figure 5, we annotate all the poisons with black star marks, where the black color does not mean a specific class. For the example case in Figure 5, the labels of poisons are from 1 to 9.
>
> (cont'd)

---

> > ### Author Response · Authors · 2022-08-02
> > **Response to Reviewer gUo3 (cont'd)**
> >
> > ***C4: Why such a big difference between the dirty-label and clean-label attacks?***
> >
> > ***R4:***
> > The difference between the dirty-label and clean-label attacks mainly comes from the data type casting in our experiments.
> >
> > Although we optimize Equation (5) in the continuous space, our poisoning samples are actually images consisting of unsigned integers. To simulate real-world attacks, in our experiments, we first save the optimized poisons into image files (floating point numbers in [0,1] $\rightarrow$ unsigned integers in [0, 255]). Then, in the training stage, we read all training images (both the clean and the malicious images) and normalize them into floating point numbers ranging from [0,1], as required by the pre-trained feature extractors. As a result, the above procedures induce two type casting operations (floating point numbers in [0,1] $\rightarrow$ unsigned integers in [0,255] $\rightarrow$ floating point numbers in [0,1]), which have introduced unintended distortions to the poisons and might cause some fluctuations in the feature space.
> >
> > We also would like to elaborate more on the difference between dirty-label and clean-label poisoning attacks *when the feature extractor gets fine-tuned* (Figure 5(c) and Figure 5(d)):
> >
> > - For the case in Figure 5(c), the dirty-label poisoning poisons are images from the target class 0 (“airplane”). As they all carry visual features from the airplane, they are hard to discriminate by the feature extractor even with fine-tuning. Consequently, in Figure 5(c), we can see that the poisons (black star marks) concentrate on class 0, but these images are labeled to the wrong classes (from 1 to 9). These points become “outliers” to increase the exposure of normal samples in class 0.
> > - For the case in Figure 5(d), the clean-label poisons are images from their original class (from 1 to 9) but with carefully crafted perturbations (we recommend observing Figure 3 to understand the difference between dirty-label poisons and clean-label poisons). In the fine-tuning process, the model ignores the small perturbations and learns the visual features to discriminate these cleanly labeled samples. In Figure 5(d), we can see that most poisons distribute within the clusters of their correct class in the feature space, which are far from normal samples in the target class 0. As a result, the poisons can hardly affect the model behavior for the target class.
> >
> >
> > ***C5: Can you show that the dirty-label attack actually works in an "end-to-end" training scenario?***
> >
> > ***R5:***
> > Thanks for your comments and suggestions. We follow a similar end-to-end training setup by [1*], where we build CNN models with a scaled-AlexNet convolutional architecture. We set the learning rate as 1e-4. The results are listed below.
> >
> > ||| Without Poisoning |||Dirty-Label Poisoning||
> > |------|------|------|------|------|------|------|
> > ||**MI AUC**|**TPR@FPR=1%**|**Test Acc.**|**MI AUC**|**TPR@FPR=1%**|**Test Acc.**|
> > |MNIST|0.532$\pm$0.018|0.98$\pm$0.39%|0.979|0.711$\pm$0.049|3.28$\pm$1.09%|0.959$\pm$0.008|
> > |CIFAR-10|0.728$\pm$0.060|1.31$\pm$0.47%|0.607|0.908$\pm$0.029|14.26$\pm$5.48%|0.576$\pm$0.010|
> > |STL-10|0.691$\pm$0.051|2.95$\pm$2.43%|0.565|0.919$\pm$0.031|13.78$\pm$6.83%|0.527$\pm$0.012|
> > |CelebA|0.702$\pm$0.001|0.00$\pm$0.00%|0.747|0.757$\pm$0.009|3.11$\pm$0.75%|0.674$\pm$0.003|
> > |PatchCamelyon|0.666$\pm$0.016|2.07$\pm$0.35%|0.815|0.702$\pm$0.003|3.31$\pm$0.13%|0.787$\pm$0.004|
> >
> > The experimental results show that the dirty-label attack is not constrained to the transfer learning setting and works in an “end-to-end” training scenario.
> >
> > (cont'd)

---

> > > ### Author Response · Authors · 2022-08-02
> > > **Response to Reviewer gUo3 (cont'd)**
> > >
> > >
> > > ***C6: Can you show how well your MI attacks work at low false-positive rates?***
> > >
> > > ***R6:***
> > > Yes. And thanks for your suggestions. We have evaluated our MI attacks at a low FPR (1%). The results are listed below.
> > >
> > > ||Without Poisoning|Dirty-Label Poisoning|Clean-Label Poisoning|
> > > |------|------|------|------|
> > > |**MNIST**||||
> > > |Xception|1.22$\pm$0.56%|4.12$\pm$1.49%|1.50$\pm$0.72%|
> > > |InceptionV3|1.01$\pm$0.39%|3.52$\pm$1.21%|1.34$\pm$0.61%|
> > > |VGG16|1.24$\pm$0.55%|3.77$\pm$1.34%|0.94$\pm$0.35%|
> > > |ResNet50|1.22$\pm$0.62%|3.68$\pm$1.56%|1.38$\pm$0.33%|
> > > |MobileNetV2|1.04$\pm$0.49%|3.91$\pm$1.06%|1.60$\pm$0.58%|
> > > |**CIFAR-10**||||
> > > |Xception|1.21$\pm$0.55%|3.08$\pm$1.21%|2.46$\pm$0.56%|
> > > |InceptionV3|1.35$\pm$0.46%|7.28$\pm$2.83%|1.44$\pm$0.59%|
> > > |VGG16|1.12$\pm$0.40%|4.60$\pm$1.26%|2.69$\pm$0.76%|
> > > |ResNet50|1.06$\pm$0.29%|7.38$\pm$2.77%|1.73$\pm$0.77%|
> > > |MobileNetV2|1.20$\pm$0.66%|3.15$\pm$1.44%|2.03$\pm$0.63%|
> > > |**STL-10**||||
> > > |Xception|1.50$\pm$0.91%|6.83$\pm$2.92%|3.90$\pm$1.63%|
> > > |InceptionV3|1.57$\pm$0.78%|12.80$\pm$8.09%|1.60$\pm$1.44%|
> > > |VGG16|1.00$\pm$0.66%|8.07$\pm$6.36%|3.80$\pm$2.71%|
> > > |ResNet50|1.65$\pm$0.79%|11.28$\pm$10.28%|2.83$\pm$1.44%|
> > > |MobileNetV2|1.12$\pm$0.92%|4.35$\pm$3.35%|2.08$\pm$1.27%|
> > > |**CelebA**||||
> > > |Xception|1.01$\pm$0.01%|2.90$\pm$1.10%|2.61$\pm$0.69%|
> > > |InceptionV3|1.10$\pm$0.28%|2.75$\pm$0.01%|1.58$\pm$0.30%|
> > > |VGG16|1.26$\pm$0.06%|2.29$\pm$0.35%|1.37$\pm$0.17%|
> > > |ResNet50|0.89$\pm$0.09%|1.92$\pm$0.04%|1.13$\pm$0.13%|
> > > |MobileNetV2|1.21$\pm$0.11%|2.22$\pm$0.04%|1.44$\pm$0.16%|
> > > |**PatchCamelyon**||||
> > > |Xception|1.18$\pm$0.04%|2.12$\pm$0.22%|1.65$\pm$0.13%|
> > > |InceptionV3|0.94$\pm$0.10%|2.38$\pm$0.02%|1.13$\pm$0.05%|
> > > |VGG16|1.15$\pm$0.05%|1.69$\pm$0.13%|1.25$\pm$0.11%|
> > > |ResNet50|1.35$\pm$0.17%|2.25$\pm$0.29%|1.47$\pm$0.11%|
> > > |MobileNetV2|1.04$\pm$0.18%|1.79$\pm$0.07%|1.17$\pm$0.05%|
> > >
> > > Note: we consider evaluation TPR at FPR=1% to accommodate our evaluation setting. We would like to take this opportunity to explain why we use FPR1%. Concretely, for evaluations on MNIST, CIFAR-10, and STL-10, the number of members is 1,000, 1,000, and 400, respectively. If we set FPR=0.1%, it means that at most, one false positive is allowed, which is impractical in our case. Besides, Tramèr et al. [3*] can use FPR=0.1% in their evaluation due to the fact that they investigate the worst-case privacy and can have larger member/non-member samples. For instance, Tramèr et al. [3*] used 25,000 members and 25,000 non-members from CIFAR 10 to evaluate their method. For the above reasons, we therefore use FPR=1%.
> > >
> > > The experimental results indicate that our attack can effectively increase the TPR at a low FPR. We can also clearly observe that the dirty-label poisoning attack has achieved higher TPRs than the clean-label poisoning attack has.
> > >
> > > **References**
> > >
> > > [1*] Ali Shafahi et al. Poison Frogs! Targeted Clean-Label Poisoning Attacks on Neural Networks. In Annual Conference on Neural Information Processing Systems (NeurIPS), pages 6103–6113. NeurIPS, 2018.
> > >
> > > [2*] Chen Zhu et al. Transferable Clean-Label Poisoning Attacks on Deep Neural Nets. In International Conference on Machine Learning (ICML), pages 7614–7623. PMLR, 2019.
> > >
> > > [3*] Florian Tramèr et al. Truth Serum: Poisoning Machine Learning Models to Reveal Their Secrets. arXiv 2204.00032, 2022.
> > >
> > > [4*] Reza Shokri et al. In IEEE Symposium on Security and Privacy (S&P), pages 3–18. IEEE, 2017.
> > >
> > > [5*] Ahmed Salem et al. ML-Leaks: Model and Data Independent Membership Inference Attacks and Defenses on Machine Learning Models. In Network and Distributed System Security Symposium (NDSS). Internet Society, 2019.

---

> > > ### Comment · Reviewer_gUo3 · 2022-08-08
> > > **Attacks invariant to integer casting**
> > >
> > > Thanks for your detailed response.
> > >
> > > Regarding R4, this seems like something you should be able to take into account in your attack.
> > > E.g., if you use PGD, after every attack step you can do the double-casting as part of the projection operation. This should usually give you adversarial examples that are mostly invariant under these castings.
> > > It would be good to try this out (or to save the datapoints as floats directly) to get a better sense for how strong this effect is in your case.

---

> > > > ### Author Response · Authors · 2022-08-08
> > > > **Yes, attacks are invariant to integer casting!**
> > > >
> > > > Thanks for your comments.
> > > >
> > > > As suggested, we have examined the L2-norm distance between each poison $x^*$ and the corresponding base sample $x\_{base}$ in the feature space (i.e., $\|g(x^*)-g(x\_{base})\|_2$). Specifically, we examined the distance metric for the original optimization outputs and the saved image poisons, respectively (dataset: CIFAR10, feature extractor: InceptionV3, target class: 0). We report the statistical results below:
> > > >
> > > > || Original optimization outputs | Image poisons | Diff. by type casting |
> > > > |---|---|---|---|
> > > > |avg.$\pm$std.|16.944$\pm$6.833|17.117$\pm$6.824|0.173$\pm$0.144|
> > > > |min|3.393|3.548|0.009|
> > > > |1st Quartile|11.933|12.157|0.101|
> > > > |2nd Quartile|16.218|16.377|0.143|
> > > > |3rd Quartile|21.101|21.314|0.207|
> > > > |max|42.925|43.086|1.977|
> > > >
> > > > We also examined the MI attack against models poisoned by the original optimization outputs and the saved image poisons. The results are below:
> > > >
> > > > |Poisoned by original optimization outputs| | Poisoned by image poisons| |
> > > > |---|---|---|---|
> > > > |**MI AUC**|**TPR@FPR=1%**|**MI AUC**|**TPR@FPR=1%**|
> > > > |0.795|0.80%|0.789|0.80%|
> > > >
> > > > The results are consistent with your hypothesis: **the impact of type casting is nearly negligible**, which means that the conclusion in our original response was untenable. We apologize for this.
> > > >
> > > > Meanwhile, we found that **some clean-label poisons had a large L2-norm distance to the base sample**. We believe this is the root cause of the difference between the dirty-label attack and the clean-label attack. For the poisons that are farther from the base samples, they have weaker impacts on the poisoned model.
> > > >
> > > > We have performed a preliminary study to examine our hypothesis. For an attack with a poisoning budget $b\_{poison}$, we chose the first 90%$b\_{poison}$ poisons **with the smallest L2-norm optimization loss** to compose the poisoning dataset (we use 10%$b\_{poison}$ clean samples for the target class). Then, we repeated the aforementioned poisoning process but randomly chose poisons. We show the results below:
> > > >
> > > > |$b\_{poison}$|With Sorted Poisons||With Random Selected Poisons||
> > > > |---|---|---|---|---|
> > > > ||**MI AUC**|**TPR@FPR=1%**|**MI AUC**|**TPR@FPR=1%**|
> > > > |200|0.731|0.90%|0.706|1.00%|
> > > > |400|0.779|1.00%|0.722|1.60%|
> > > > |600|0.786|1.10%|0.757|1.70%|
> > > > |1000 (all included)|0.795|0.80%|0.795|0.80%|
> > > >
> > > > We can understand the results from two aspects:
> > > >
> > > > 1. Attacks with sorted poisons have stronger poisoning effects than attacks with randomly selected poisons. The reason is that for the former, the poisons are closer to the base samples in the input space.
> > > > 2. We can find that we can use 60% poisons ($b\_{poison}=600$, MI AUC=0.786) to achieve an attack performance as nearly good as the attack with 100% poisons ($b\_{poison}=1,000$, MI AUC=0.795). It is likely that 40% of poisons have slight effects on the victim model.
> > > >
> > > > We will discuss this in the main paper and add more details in the final version.
> > > >
> > > > Following this line, we think there are several potential ways to improve the clean-label attack:
> > > > 1. Design a more effective perturbation generation algorithm like [6*];
> > > > 2. Devise a strategy to select the base image for each poison to have a better match in the input space;
> > > > 3. Select "stronger" poisons when generating poisoning datasets.
> > > >
> > > > We plan to investigate the above research tasks in the future.
> > > >
> > > > Thanks again for your comments, which help to inspire more ideas. We are happy to answer if there are additional issues/questions.
> > > >
> > > > **References**
> > > >
> > > > [6*] Cassidy Laidlaw and Soheil Feizi. Functional Adversarial Attacks. In Annual Conference on Neural Information Processing Systems (NeurIPS). NeurIPS, 2019.

---

### Official Review · Reviewer_ki6P · 2022-07-06

**Rating:** 3
**Confidence:** 4
**Soundness:** 1 poor
**Presentation:** 2 fair
**Contribution:** 1 poor

**Summary:**

This paper address the problem of membership exposure, where the attacker tries to identify if a specific sample belongs to the training set. The authors propose to apply data poisoning attacks such as label flip attacks and clean-label attacks (feature collision) to amplify such membership identification. Experiments empirically verify the effectiveness of such attacks.

**Questions:**

(1) In Equation (2), the authors propose to examine if a point $(x,y)$ belongs to the training set using a classification metric (cross-entropy loss). This metric does not make sense to me as it only determines if the point is an outlier of the distribution. Even if $M_{mem}$ is high, it only reveals $(x,y)$ is not an outlier, but does not guarantee $(x,y)$ is in D_train. I expect the authors to explain more about this.

(2) In Section 4.1, the authors split the datasets evenly into D_clean, D_test, and D_shadow. Although this assumption ensures that D_clean and D_shadow come from the same dataset, this significantly reduces the size of D_clean and might make it easier to poison. Can you examine this by making $\|\mathcal{D}_{shadow}\|$ smaller?

(3) Additionally, It seems that the poisoning budget is much smaller than $\|\mathcal{D}_{shadow}\|$, then do we need to have such a big D_shadow?

**Limitations:**

Limitations are addressed in this paper.

**Strengths And Weaknesses:**

Strengths:

(1) This paper proposes that data poisoning attacks might help amplify membership exposure, which is novel.

Weakness:

(1) The author states in line 156 that "The key to amplifying membership exposure is to cause overfitting in the targeted class." However, I don't think this is necessarily true. From my understanding, the authors try to poison Equation (2) by making $f(\cdot)$ less accurate using simple poisoning attacks such as label flip attacks. However, the author does not show if degrading the performance of $f(\cdot)$ is related to the MI AUC. Thus why this attack is effective is not clear to me.

(2) The methods proposed in this paper (label flip attacks, and clean label attacks using label flip) do not provide me insights into solving the MI problem and Equation (2), as the objective is not considered in the approaches which make them heuristic.

(3) The experimental section is very confusing to me. From the content, the MI attack aims to identify if a point $(x,y)$ belongs to the training set, but Table 1 shows the MI AUC of the target class.

(4) Why is the baseline AUC so high? Looks like without poisoning the MI AUC can already become as high as 74.8%, which is itself shocking to me. Is this problem really so serious or is it illy defined?

(5) See my questions for more concerns.

---

> ### Author Response · Authors · 2022-08-02
> **Response to Reviewer ki6P**
>
> We thank the reviewer for their valuable comments (C). We hope that our responses (R) have fully addressed all of the reviewer’s concerns, and remain committed to clarifying any further questions that may arise during the discussion period.
>
> ***C1: The author states in line 156 that "The key to amplifying membership exposure is to cause overfitting in the targeted class." However, I don't think...***
>
> ***R1:***
> Thanks for your comments (we are afraid that “Equation (2)” in the original comments should be “Equation (1)?”)
>
> To address the reviewer’s concern and help the reviewer better understand how our attacks work, we compare the testing accuracy on the target class before/after poisoning.
>
> ||Without Poisoning|Dirty-Label Poisoning|Clean-Label Poisoning|
> |------|------|------|------|
> |**MNIST**||||
> |Xception|0.940$\pm$0.025|0.785$\pm$0.071|0.815$\pm$0.081|
> |InceptionV3|0.929$\pm$0.031|0.748$\pm$0.063|0.785$\pm$0.087|
> |VGG16|0.954$\pm$0.017|0.805$\pm$0.060|0.887$\pm$0.043|
> |ResNet50|0.967$\pm$0.014|0.802$\pm$0.108|0.941$\pm$0.019|
> |MobileNetV2|0.961$\pm$0.021|0.769$\pm$0.063|0.871$\pm$0.070|
> |**CIFAR-10**||||
> |Xception|0.769$\pm$0.084|0.567$\pm$0.074|0.573$\pm$0.090|
> |InceptionV3|0.677$\pm$0.078|0.519$\pm$0.048|0.591$\pm$0.067|
> |VGG16|0.815$\pm$0.068|0.586$\pm$0.043|0.609$\pm$0.055|
> |ResNet50|0.849$\pm$0.058|0.691$\pm$0.057|0.721$\pm$0.063|
> |MobileNetV2|0.843$\pm$0.060|0.647$\pm$0.048|0.675$\pm$0.057|
> |**STL-10**||||
> |Xception|0.858$\pm$0.050|0.689$\pm$0.053|0.677$\pm$0.069|
> |InceptionV3|0.759$\pm$0.083|0.592$\pm$0.061|0.666$\pm$0.083|
> |VGG16|0.876$\pm$0.050|0.645$\pm$0.045|0.670$\pm$0.080|
> |ResNet50|0.898$\pm$0.038|0.702$\pm$0.034|0.768$\pm$0.051|
> |MobileNetV2|0.935$\pm$0.028|0.719$\pm$0.048|0.781$\pm$0.048|
> |**CelebA**||||
> |Xception|0.725$\pm$0.003|0.702$\pm$0.036|0.658$\pm$0.065|
> |InceptionV3|0.688$\pm$0.041|0.642$\pm$0.018|0.645$\pm$0.026|
> |VGG16|0.725$\pm$0.005|0.673$\pm$0.026|0.657$\pm$0.016|
> |ResNet50|0.744$\pm$0.075|0.662$\pm$0.050|0.756$\pm$0.014|
> |MobileNetV2|0.750$\pm$0.008|0.667$\pm$0.016|0.634$\pm$0.030|
> |**PatchCamelyon**||||
> |Xception|0.847$\pm$0.001|0.731$\pm$0.003|0.798$\pm$0.004|
> |InceptionV3|0.833$\pm$0.001|0.713$\pm$0.046|0.801$\pm$0.022|
> |VGG16|0.862$\pm$0.000|0.809$\pm$0.019|0.821$\pm$0.017|
> |ResNet50|0.892$\pm$0.013|0.714$\pm$0.049|0.847$\pm$0.003|
> |MobileNetV2|0.890$\pm$0.022|0.719$\pm$0.002|0.797$\pm$0.017|
>
>
> It can be seen that our proposed attacks have significantly decreased the testing accuracy on the target class, causing a larger gap between the training accuracy and testing accuracy. In other words, our attacks cause severer overfitting on the target class. As discussed by many previous references, severer overfitting would likely cause more significant membership exposure [1*, 2*]. Our evaluation results are consistent with these conclusions.
>
> ***C2: The methods proposed in this paper (label flip attacks, and clean label attacks using label flip) do not provide me insights into solving the MI problem and Equation (2), as the objective is not considered in the approaches which make them heuristic.***
>
> ***R2:***
> First, we would like to explain why we chose Equation (2) in our MI problem.
> Note that $-(1-x)\\log(x)$ is monotonically decreasing and $-\log(1-x)x$ is monotonically increasing.
> As a result, if $f(x)_{y}$ (the confidence score of the correct label $y$) decreases, the first item of Equation (2), $-(1-f(x)_y)\\log(f(x)_y)$ will increase.
> Meanwhile, if the confidence score of other class $f(x)_i$ increases, $-\\log(1-f(x)_i)f(x)_i$ will increase.
> There are two extreme cases:
>
> - $f(x)_y=1$. The model has 100% prediction confidence on the correct label and $M_\{mem\}=0$.
> - $f(x)_i=1, i \neq y$. The model has 100% prediction confidence on an incorrect label and  $M_\{mem\}=\infty$.
>
> With the above properties, Equation (2) can be a good indicator of the correctness of the model prediction. The higher confidence the model predicts on the correct label, the lower is $M_{mem}$. Meanwhile, the higher confidence the model predicts on an incorrect label, the higher is $M_{mem}$. The explanations of the design of Equation (2) can also be found in [3*, 4*]. In general, a model will have higher confidence to predict the correct label on training samples and hence have lower $M_{mem}$.
>
> Then, we would like to explain why Equation (2) is not considered in the objective. The main reason is that calculating Equation (2) requires full access to the target model and the clean dataset $D_{train}$ during the training process, which is unrealistic. Moreover, even $D_{train}$ and the target model are accessible, the crafting of poisons suffers a high computational cost [5*]. Instead, we choose to exploit the classical label-flipping attacks to decrease the model accuracy on testing samples (i.e., the non-members), which can be detected by Equation (2).
>
> (cont'd)

---

> > ### Author Response · Authors · 2022-08-02
> > **Response to Reviewer ki6P (cont'd)**
> >
> > ***C3: The experimental section is very confusing to me. From the content, the MI attack aims to identify if a point $(x,y)$  belongs to the training set, but Table 1 shows the MI AUC of the target class.***
> >
> > ***R3:***
> > Thanks for your comments. First, we would like to demonstrate how our MI attack works. As discussed in R2, we use Equation (2) as an indicator to discriminate whether a sample $(x,y)$ belongs to the training set. For sample $(x,y)$, if $M_{mem}$ is below a predefined threshold $\tau$, we think the sample is a member; otherwise, we think the sample is a non-member. In our experiments, we run the MI attack on each sample in the evaluation dataset and compute the true positive rate (TPR) and false positive rate (FPR), where we regard being a member as a positive event. To gain a holistic view of our attack performance, we evaluate the MI accuracy by varying $\tau$, plot the ROC curve (Figure 2 is one example), and compute the AUC (area under the curve) under the ROC curve as our evaluation metric. The higher the AUC score is, the higher the risk of membership exposure is since we can find a threshold $\tau$ with high TPR and low FPR in this case. Such a threshold corresponds to an accurate MI attacker. We will clarify this in our revision.
> >
> > ***C4: Why is the baseline AUC so high? Looks like without poisoning the MI AUC can already become as high as 74.8%, which is itself shocking to me. Is this problem really so serious or is it illy defined?***
> >
> > ***R4:***
> > For the baselines in our experiment, we use the basic transfer learning setup without regulation or other strategies to improve generalization. So there are chances that the model is more significantly overfitting to the training data and leads to a high MI attack AUC. The high-AUC baselines in our experiments reveal that membership exposure is easy to happen if the hyper-parameters and student models are not carefully configured. We can also see that not all baselines have high AUCs. For instance, with the CelebA dataset, although the average AUC for InceptionV3 baseline is 0.748, the average AUC for ResNet50 is 0.571.
> >
> > In the end, we would like to clarify that the goal of this work is not to achieve an absolute high MI attack accuracy. Instead, we show our attacks can increase the membership exposure risks. To this end, we should pay more attention to the MI attack AUC change before and after our poisoning attacks.
> >
> >
> > ***C5: In Equation (2), the authors propose to examine if a point $(x,y)$ belongs to the training set using a classification metric (cross-entropy loss)...***
> >
> > ***R5:***
> > Thanks for your comments.
> > First, we would like to clarify that a low $M_{mem}$ indicates $(x,y)$ has less chance of being an outlier (please see R2 for illustrations).
> >
> > The intuition of our MI attack is that the model tends to have less cross-entropy loss on the training samples, as this is what the training process does. Based on this intuition, we use $M_{mem}$ as an indicator to infer whether a sample $(x,y)$ is in the training dataset. If $M_{mem}$ is lower than a predefined threshold, we believe $(x,y)$ has a better fitting on the training set.
> >
> > In an ideal setting, a good model can well learn the data distribution and have the same performance on the training samples and testing samples. But there always exists a generalization gap for the model, i.e., the model behaves differently on training samples and testing samples. This gap gives us clues to infer whether a sample is in the training dataset [1*]. We agree that a low $M_{mem}$ does not guarantee $(x,y)$ is in $D_{train}$, but in most cases, training samples tend to have lower $M_{mem}$.
> >
> > (cont'd)

---

> > > ### Author Response · Authors · 2022-08-02
> > > **Response to Reviewer ki6P (cont'd)**
> > >
> > > ***C6: In Section 4.1, the authors split the datasets evenly into $\mathcal{D}_{clean}$, $\mathcal{D}_{test}$, and $\mathcal{D}_\{shadow\}$...***
> > >
> > > ***R6:***
> > > Thanks for your question. We study the impact of the number of poisons in Section 5.2 in our submitted paper. We gradually increase the number of poisons from 50 to 1,000, which simulates $\|\mathcal{D}_\{shadow\}\|=[5\%, 100\%]\mathcal{D}_\{clean\}$. We report the results in Figure 4. The results are not surprising: the MI AUC increases as $\|\mathcal{D}_\{shadow\}\|$ increases, where we inject more poisons into the clean dataset. Take ResNet50 transfer learning models on the CIFAR-10 dataset for instance  (baseline AUC without poisoning: 0.597), when $\|\mathcal{D}_\{shadow\}\|=10\%\mathcal{D}_\{clean\}$, the membership inference AUC score is 0.7076 on average, and when $\|\mathcal{D}_\{shadow\}\|=\mathcal{D}_\{clean\}$, the AUC is 0.93 on average.
> > >
> > > ***C7: Additionally, It seems that the poisoning budget is much smaller than $\|\mathcal{D}_\{shadow\}\|$, then do we need to have such a big $\mathcal{D}_\{shadow\}$?***
> > >
> > > ***R7:***
> > > We do not have such a big $\mathcal{D}_\{shadow\}$.
> > > We split the dataset by letting $\|\mathcal{D}_\{shadow\}\|=\|\mathcal{D}_\{clean\}\|$ just for the convenience of evaluation.
> > > In practice, the attacker just needs to collect $b_\{poison\}$ samples from the target class.
> > > For clean-label poisoning attacks, the attacker also needs $\frac{(c-1)b_\{poison\}}{c}$ samples from other classes, where $c$ refers to the number of classes for the victim model.
> > >
> > > **References**
> > >
> > > [1*] Reza Shokri et al. Membership Inference Attacks Against Machine Learning Models. In IEEE Symposium on Security and Privacy (S&P), pages 3–18. IEEE, 2017.
> > >
> > > [2*] Ahmed Salem et al. ML-Leaks: Model and Data Independent Membership Inference Attacks and Defenses on Machine Learning Models. In Network and Distributed System Security Symposium (NDSS). Internet Society, 2019.
> > >
> > > [3*] Liwei Song et al. Privacy Risks of Securing Machine Learning Models against Adversarial Examples. In ACM SIGSAC Conference on Computer and Communications Security (CCS), pages 241–257. ACM, 2019.
> > >
> > > [4*] Liwei Song and Prateek Mittal. Systematic Evaluation of Privacy Risks of Machine Learning Models. In USENIX Security Symposium (USENIX Security), pages 2615–2632. USENIX, 2021.
> > >
> > > [5*] W. Ronny Huang et al. MetaPoison: Practical General-purpose Clean-label Data Poisoning. In Annual Conference on Neural Information Processing Systems (NeurIPS), pages 12080–12091. NeurIPS, 2020.

---

### Official Review · Reviewer_Ewj4 · 2022-07-08

**Rating:** 3
**Confidence:** 5
**Ethics Flag:** Yes
**Soundness:** 3 good
**Presentation:** 3 good
**Contribution:** 1 poor

**Summary:**

This paper presents data poisoning attacks for increasing the risk of membership inference (a type of privacy attack on machine learning models). The paper studies two different strategies: dirty-label attacks (i.e., label-flipping attacks) and clean-label attacks. In evaluation with image classification datasets, the paper shows that dirty-label attacks can increase the average attack success (represented by the AUC metric) by ~0.25 compared to the clean baselines with negligible degradation of accuracy.

**Questions:**

1. Scientific comparison to the prior work [1, 2] and clarification of the scope of the contributions.
2. Clarification on the contributions of this paper considering research questions in the community.
    1. RQ 1. Why does this poisoning work?
    2. RQ 1. What are the training samples that become more vulnerable after poisoning?
    3. RQ 2. What does it mean to increase the AUC?
    4. RQ 3. Why are clean-label poisoning attacks important?
    5. RQ 4. If someone wants to mitigate this attack, what can this person do?
3. Clarification of the interpretation of the results (Questions from 1 to 4 above).

**Ethics Review Area:**

["Privacy and Security (e.g., consent)"]

**Limitations:**

My lesser concern is that the paper conducts offensive research, contaminating the training data for increasing privacy risks but does not discuss any ethical concerns when a miscreant uses this attack. At least, I think it's okay to talk about some potential defense mechanisms, but the paper just concludes that defenses are future work.

**Strengths And Weaknesses:**

**Strengths**
1. The paper studies an interesting problem.
2. The paper is well-written and easy to read.


**Weaknesses**
1. Prior work has already studied the claimed contributions.
2. Poor comparison with the literature on accessing privacy risks.
3. Weak evaluations.


**Detailed Comments**

The idea of evaluating the risk of membership inference under data poisoning attacks is interesting. As more and more data is collected from various sources, the privacy risks of machine learning models trained on such data is an important topic.

**1. Contributions were shown by the prior work**

However, data poisoning for increasing privacy risks has already been initially studied by Mahloujifar et al. [1], and all the contributions (claimed from Line 41 to Line 52) have already been shown by Tramer et al. [2]. Moreover, the paper uses the techniques and tools for measuring the membership inference risks already known as meaningless by Carlini et al. [3]. Thus, I believe this paper is largely detached from the state-of-the-art privacy studies, and unfortunately, the contributions are the repetition of what we have known so far.

[1] Mahloujifar, et al., Property Inference from Poisoning, IEEE Security and Privacy, 2022.
[2] Tramer et al., Truth Serum: Poisoning Machine Learning Models to Reveal Their Secrets, Preprint, 2022.
[3] Carlini et al., Membership Inference Attacks From First Principles, IEEE Security and Privacy, 2022.

**Note:** The studies I mentioned had appeared 3-12 months before the NeurIPS submission deadline and were even accepted before then, so I wouldn’t review this paper as concurrent work.


**2. Poor comparison to the prior work**

My second concern is that the paper just combines two threat models (data poisoning and membership inference attacks) while it largely ignores important research questions in the community, such as:

RQ 1. Why does this poisoning work?
RQ 1. What are the training samples that become more vulnerable after poisoning?
RQ 2. What does it mean to increase the AUC?
RQ 3. Why are clean-label poisoning attacks important?
 (As the paper mentioned in the introduction, sanitizing the training data is not feasible.)
RQ 4. If someone wants to mitigate this attack, what can this person do?

which (those questions) are partially already answered in the prior work [2, 3].


**3. Weak Evaluation**

My last concern is that there is unclear interpretation of the results in the evaluation section:

Q1. (Line 257) I am unclear why the clean-label poisoning attack can be considered an “approximate” version of the dirty-label poisoning attack in the feature space?

As shown in visualization (Figure 5), it seems that clean-label attacks and dirty-label attacks cause a completely different impact on the models. If this is true, wouldn’t it make more sense in Sec 3 to present a single attack with different objectives?

Q2. (Line 261) I am also unclear how this paper measures the distributional differences between D_train and D_shadow. I believe it’s still a hard question to quantify the distributional differences and actively studied in domain adaption and robustness, so I don’t think we can compare.

Q3. (Line 284) I am a bit confused about the fine-tuning scenario. Is it the case where we take an ImageNet pre-trained model and fine-tune it on CIFAR10? Then why don’t the attacker make membership inference on ImageNet instead of attacking CIFAR10? Isn’t it easier to spot poisoning samples if we inject them into the training data for fine-tuning?

Q4. (Line 304) I am unclear about the connection between the presented attacks and adversarial training. Adversarial training crafts adversarial examples in each training iteration and update the model parameters, while this attack just injects a set of static poisons into the training data and trains on it.

---

> ### Author Response · Authors · 2022-08-02
> **Response to Reviewer Ewj4**
>
> We thank the reviewer for the valuable comments (C). We hope that our responses (R) have fully addressed all of the reviewer’s concerns, and remain committed to clarifying any further questions that may arise during the discussion period.
>
> ***C1-1: Scientific comparison to the prior work [1, 2] and clarification of the scope of the contributions.***
>
> ***R1-1:***
> Thanks for your comments. Here we clarify the difference between our work with the mentioned two, respectively.
>
> Note that we map your reference [1], [2] and [3] to [1*], [2*] and [3*] respectively so that we can offer a consistent reference scheme in the rebuttal.
>
> __Differences between ours and Mahloujifar et al. [1*]:__
>
> Mahloujifar et al. [1*] mainly study the property inference problem, while our work investigates the membership inference attack problem.
>
> __Differences between ours and Tramèr et al. [2*]:__
>
> In addition to the comments by gUo3, there are differences between our attack with the concurrent work of Tramèr et al. [2*].
>
> - Our proposed attack mainly aims at the untargeted attack, while the concurrent work mainly aims at the targeted attack, where the latter requires the attack to determine which sample $(x,y)$ to infer in advance.
>
> - Besides the classic label-flipping poisoning attack [4*], we also investigate the feasibility of clean-label poisoning. With clean-label poisoning attacks, the attacker can craft more natural and stealthy poisoning samples. We think our work can be a good complement to existing clean-label poisoning attacks that try to cause misclassification.
>
> In summary, our contributions mainly include:
> We reveal there is an underexplored threat of data poisoning, by demonstrating the feasibility of utilizing data poisoning attacks to amply membership exposure.
> Inspired by the classic label-flipping method [4*], we design a robust dirty-label poisoning attack, which is effective in the end-to-end learning scenario.
> Moreover, we design a stealthy clean-label poisoning attack in the transfer learning setting, which is more practical without the control of the labeling process.
>
> ***C1-2: Moreover, the paper uses the techniques and tools for measuring...***
>
> ***R1-2:***
> Thanks for your comments. We did not use the attack proposed Carlini et al. [3*] as it is based on a stronger assumption of the attacker, and the computing costs are much higher than the attack we used. Please refer to **R2** in the response to **Reviewer 47fC** for more details. Note that we agree that the worst-case privacy investigated by Carlini et al. [3*] cannot be effectively evaluated by the average-case methodology. However, we argue that privacy should be looked into on a case-by-case basis. For instance, the same authors (Carlini and Tramèr et al.) used the average-case methodology to evaluate the privacy risk of training data extraction from large-scale language models [15*].
>
> Besides, as suggested by other reviewers and Carlini et al. [3*], we also examine the worst case when FPR is low. Please refer to **R6** in the response to **Reviewer gUo3** for more details.
>
> ---
> **Clarification on the contributions of this paper considering research questions (RQ1~RQ4 in the original comments) in the community.**
>
> ***C2 (RQ1 in the original comments): Why does this poisoning work?***
>
> ***R2:***
> The intuition of our attack is to worsen the overfitting on the target class, as most prior arts point out that overfitting is a primary reason for membership exposure [5*, 6*]. To this end, we propose to exploit the classic label-flipping poisoning method to generate “outliers” around the normal samples to degrade the testing accuracy on the target class. Since existing neural networks have a large capacity, they learn both normal samples and outliers simultaneously [16*], if no defensive strategies like regularization are considered. Consequently, the poisoned model will get overfitted to the normal samples and poison samples (we provide Figure 1 in our submitted paper to help readers understand this phenomenon). Then, the attacker can infer whether a sample belongs to the training samples of the target class, by analyzing the model behavior difference, i.e., the overfitting problem, on the training samples and testing samples.
>
> We have designed an evaluation metric to help quantitatively understand the membership exposure phenomenon in our experiments. We reported our empirical study results in Section B of the submitted appendix.
>
> ***C3 (RQ1 in the original comments): What are the training samples that become more vulnerable after poisoning?***
>
> ***R3:***
> Thanks for your questions. In Section B of the submitted appendix, we find that for a model, when its training samples are closer to samples from other classes in the input space, the model will suffer from a higher chance of membership exposure. Using this observation, the training samples that are closer to the poisoning samples will be more vulnerable in the input space.
>
> (cont'd)

---

> > ### Author Response · Authors · 2022-08-02
> > **Response to Reviewer Ewj4 (cont'd)**
> >
> > ***C4 (RQ2 in the original comments): What does it mean to increase the AUC?***
> >
> > ***R4:***
> > For our evaluation scheme, a higher AUC means a stronger attack to discriminate between the member and non-members. We will clarify this in the revision. Below is more detailed explanations:
> >
> > Our membership inference attack leverages the *score-based* attack scheme. With our adopted metric $M_{mem}$ shown in Equation (2), we compare it with a predefined threshold, if $M_{mem}$ is below a predefined threshold $\tau$, we consider the sample is a member; otherwise, we treat the sample as a non-member.
> >
> > By changing the threshold, we can compute different true positive rate (TPR) and false positive rate (FPR), where we regard being a member as a positive event. We plot the ROC curve (see Figure 2 in the original submission) and compute the area under the curve (AUC). AUC refers to the probability that a two-class classifier will discriminate between a randomly selected positive instance and a negative instance. Indeed, AUC has been widely adopted for evaluating two-class classifiers [7*, 8*], and a high AUC means a better classification performance. In our work, a high AUC means the attacker is strong to discriminate between the members and non-members.
> >
> > Besides, as suggested by other reviewers, we also examine the worst case when FPR is low. Please refer to **R6** in the response to **Reviewer gUo3** for more details.
> >
> > ***C5 (RQ3 in the original comments): Why are clean-label poisoning attacks important?***
> >
> > ***R5:***
> > Compared to dirty-label poisoning attacks, clean-label poisoning attacks have two advantages:
> >
> > Clean-label poisoning attacks do not require the control of the labeling process. In this case, the poisoning samples are correctly labeled by some certified sectors.
> > Clean-label poisoning attacks are more stealthy, as the poisons have natural appearances and clean labels, making such attacks easy to evade moderation.
> >
> > In short, clean-label poisoning attacks pose more practical threats. Over recent years, clean-label poisoning attacks have attracted more and more attention in the research community [9*, 10*]. But existing attacks mainly focus on causing test-time accuracy degradation or controlled mispredictions. In our work, we aim to reveal a new threat by poisoning attacks, wherein an attacker can amplify the membership exposure of normal training samples by injecting malicious samples. Our results indicate that there is a connection between two major concerns of machine learning: data integrity and data confidentiality. We hope our attacks can complement prior research efforts on poisoning attacks.
> >
> > We will clarify this in the revision.
> >
> >
> > ***C6 (RQ4 in the original comments): If someone wants to mitigate this attack, what can this person do?***
> >
> > ***R6:***
> > The person can mitigate this attack with existing privacy-enhancement or overfitting-prevention strategies, such as differentially private training, early stopping, and normalization.
> >
> > In Section C of the submitted appendix, we show that differentially private training techniques can significantly reduce the negative impacts of poisons. We have also evaluated other standard defensive strategies. Please refer to **R4** in the response to **Reviewer 47fC** for more details.
> >
> > (cont'd)

---

> > > ### Author Response · Authors · 2022-08-02
> > > **Response to Reviewer Ewj4 (cont'd)**
> > >
> > > **Clarification of the interpretation of the results (Questions from 1 to 4 in the original comments about the evaluation results).**
> > >
> > > ***C7 (Q1 on the results in the original comments)***
> > >
> > > ***C7-1: (Line 257) I am unclear why the clean-label poisoning attack can be considered an “approximate” version of the dirty-label poisoning attack in the feature space?***
> > >
> > > ***R7-1:***
> > > Thanks for your question.
> > >
> > > First, we would like to explain the relationship between our clean-label attack and dirty-label attack. We sincerely apologize for the typo in Equation (3). The constraint should be
> > >
> > > $$
> > > \text{s.t.} \|x’-x\|_{\infty} \leq \epsilon, x’ \in \mathcal{X}
> > > $$
> > >
> > >
> > > (You can also refer to our response **Typo in Equation (3)**). After solving the optimization problem described in Equation (3), the poisoning sample $(x^*,y)$ has the same latent feature as a sample in the target class (e.g., _airplane_). Still, it has an appearance of the labeled class (e.g., _cat_). We recommend checking the example in Figure 3: The first clean-poisoning example on the right has a _cat_ appearance and label, but it has an _airplane_ latent feature.
> > >
> > > Note that, in the transfer learning case (without fine-tuning), the training process is essentially to train the newly added layers on $\{ (g(x),y) \}, (x,y) \in \mathcal{D}_{train}$. From our above example, we will train the model on an image with _airplane_ latent feature but labeled as _cat_. From this point of view, our cleaning-label attack actually crafts a set of dirty-label poisons in the input space.
> > >
> > > Then, we would like to explain why we think the clean-label poisoning attack is an “approximate” version. We conduct the optimization with continuous numbers. However, in practice, we need to convert the results into real image files, which are represented by unsigned integers. Such typecasting would introduce distortion to the poisons and harm the poisoning effectiveness. Please refer to **R5** in the response to **Reviewer gUo3** for more details.
> > >
> > >
> > > ***C7-2: As shown in visualization (Figure 5), it seems that clean-label attacks and dirty-label attacks cause a completely different impact on the models…***
> > >
> > > ***R7-2:***
> > > Thanks for your comments. The difference is caused by the fine-tuning process. (Please refer to **R4** in the response to **Reviewer gUo3**.)
> > >
> > > We thank the reviewer for suggestions for enhancing the robustness of clean-label poisoning attacks. Designing clean-label poisoning attacks in the “end-to-end” setting, such as [11*], may be possible. But it requires a more complicated design, and the optimization process induces much more computational costs. Our primary goal is to show the possibility of amplifying membership exposure with poisoning attacks. Finding more robust clean-label poisoning attacks is good to have but out of our main scope. We plan to investigate this direction in the future.
> > >
> > >
> > > ***C8 (Q2 on the results in the original comments): (Line 261) I am also unclear how this paper measures the distributional differences between D_train and D_shadow....***
> > >
> > > ***R8:***
> > > Thanks for pointing out this issue, and we agree that it is hard to quantify the distribution between the two datasets.
> > >
> > > We are sorry for the less precise expression here. We have visualized the features of samples in $D_{train}$ and $D_{shadow}$ from the studied class (0 and 3) with t-SNE techniques, to help readers qualitatively understand the sample distribution. We find that if $D_{shadow}$ comes from CIFAR-10, the clusters of $D_{shadow}$ and $D_{train}$ are nearly overlapped. On the contrary, when $D_{shadow}$ comes from STL-10, the clusters of $D_{shadow}$ and $D_{clean}$ are less overlapped.
> > >
> > > We will revise the content in Section 5.1 and add the visualization results in the revision.
> > >
> > > (cont'd)

---

> > > > ### Author Response · Authors · 2022-08-02
> > > > **Response to Reviewer Ewj4 (cont'd)**
> > > >
> > > > ***C9 (Q3 on the results in the original comments): (Line 284) I am a bit confused about the fine-tuning scenario…***
> > > >
> > > > ***R9:***
> > > > Yes, it is the case where we take an ImageNet pre-trained model and fine-tune it on CIFAR-10. We do not consider membership inference on ImageNet, but on the student dataset for the following reasons:
> > > >
> > > > 1. Most developers of public pre-trained models choose to release the training dataset information. In this case, the membership information of training samples is public, and the training data privacy is not a primary concern of the model developer.
> > > > 2. It is a common practice to fine-tune a public pre-trained model on a small proprietary student dataset, such as medical images [12*], where the student dataset privacy is often a major concern.
> > > > 3. The functionality of the fine-tuned model has been tailored to the student dataset task, i.e., CIFAR-10 classification. The model outputs, which have ten dimensions corresponding to the ten categories in CIFAR-10, will reflect more information of the CIFAR-10 data instead of the ImageNet data. So technically, it is easier to infer the membership information of CIFAR-10 samples.
> > > >
> > > > We consider the fine-tuning scenario mainly because, like prior clean-label poisoning attacks [9*, 10*], we craft poisons on the pre-trained weights of the feature extractor (as shown by Equation (3)). However, fine-tuning will change the weights of the feature extractor, which might make our attack ineffective. So we believe it is necessary to investigate the robustness of the clean-label poisons against weight fine-tuning. Our results in Section 5.3 show that clean-poisoning attacks are far less effective in the fine-tuning scenario.
> > > >
> > > > However, our dirty-label poisoning attacks are still effective in the fine-tuning setting, as the attack objective of our dirty-label attacks is independent of the weights of the feature extractors. Indeed, our dirty-label poisoning attacks work well in end-to-end scenarios. Please refer to **R5** in the response to **Reviewer gUo3** for more details.
> > > >
> > > > ***C10 (Q4 on the results in the original comments): (Line 304) I am unclear about the connection between the presented attacks and adversarial training...***
> > > >
> > > > ***R10:***
> > > > To the best of our knowledge, the adversarial training techniques to improve the model robustness contain two categories:
> > > > One is to generate adversarial examples in each training iteration or add the adversarial attack objective directly to the learning objective;
> > > > While the other is to generate static poisons adversarial examples before the training begins [13*]. In this case, we can see the adversarial examples compose a kind of augmentation dataset to the clean dataset [14*]. Our poisoning attack belongs to this category.
> > > >
> > > > Thanks for pointing this issue out. We will make this point clear in the revision.
> > > >
> > > > ***C11: My lesser concern is that the paper conducts offensive research, contaminating the training data for increasing privacy risks...***
> > > >
> > > > ***R11:***
> > > > Thanks for your comments. We have shown that DP-SGD can effectively defend our attack in Section C of the appendix. We have also evaluated standard defensive strategies. Please refer to **R4** in the response to **Reviewer 47fC** for more details.
> > > >
> > > > (cont'd)

---

> > > > > ### Author Response · Authors · 2022-08-02
> > > > > **Response to Reviewer Ewj4 (references)**
> > > > >
> > > > > **References**
> > > > >
> > > > > [1*] Saeed Mahloujifar et al. Property Inference from Poisoning. In IEEE Symposium on Security and Privacy (S&P). IEEE, 2022.
> > > > >
> > > > > [2*] Florian Tramèr et al. Truth Serum: Poisoning Machine Learning Models to Reveal Their Secrets. arXiv 2204.00032, 2022.
> > > > >
> > > > > [3*] Nicholas Carlini et al. Membership Inference Attacks From First Principles. In IEEE Symposium on Security and Privacy (S&P). IEEE, 2022.
> > > > >
> > > > > [4*] Battista Biggio et al. Poisoning Attacks against Support Vector Machines. In International Conference on Machine Learning (ICML). icml.cc / Omnipress, 2012.
> > > > >
> > > > > [5*] Samuel Yeom et al. Privacy risk in machine learning: Analyzing the connection to overfitting. In 2018 IEEE 31st Computer Security Foundations Symposium (CSF), pages 268–282. IEEE, 2018.
> > > > >
> > > > > [6*] Reza Shokri et al. In IEEE Symposium on Security and Privacy (S&P), pages 3–18. IEEE, 2017.
> > > > >
> > > > > [7*] Matthew Fredrikson et al. Privacy in Pharmacogenetics: An End-to-End Case Study of Personalized Warfarin Dosing. In dthe 2014 USENIX Security Symposium (USENIX Security). USENIX, 2014.
> > > > >
> > > > > [8*] Apostolos Pyrgelis et al. Knock Knock, Who’s There? Membership Inference on Aggregate Location Data. In the 2018 Network and Distributed System Security Symposium (NDSS). Internet Society, 2018.
> > > > >
> > > > > [9*] Ali Shafahi et al. Poison Frogs! Targeted Clean-Label Poisoning Attacks on Neural Networks. In Annual Conference on Neural Information Processing Systems (NeurIPS), pages 6103–6113. NeurIPS, 2018.
> > > > >
> > > > > [10*] Chen Zhu et al. Transferable Clean-Label Poisoning Attacks on Deep Neural Nets. In International Conference on Machine Learning (ICML), pages 7614–7623. PMLR, 2019.
> > > > >
> > > > > [11*] W. Ronny Huang et al. MetaPoison: Practical General-purpose Clean-label Data Poisoning. In Annual Conference on Neural Information Processing Systems (NeurIPS), pages 12080–12091. NeurIPS, 2020.
> > > > >
> > > > > [12*] Maithra Raghu et al. Transfusion: Understanding Transfer Learning for Medical Imaging. In Annual Conference on Neural Information Processing Systems (NeurIPS), NeurIPS, 2019.
> > > > >
> > > > > [13*] Ian Goodfellow et al. Explaining and Harnessing Adversarial Examples. In International Conference on Learning Representations (ICLR), ICLR, 2015.
> > > > >
> > > > > [14*] Christian Szegedy et al. Intriguing properties of neural networks. arXiv 1312.6199v4, 2013.
> > > > >
> > > > > [15*] Carlini, Nicholas, et al. Extracting training data from large language models. In the 30th USENIX Security Symposium (USENIX Security 21). 2021.
> > > > >
> > > > > [16*] Manoj, Naren, and Avrim Blum. Excess capacity and backdoor poisoning. In Annual Conference on Neural Information Processing Systems (NeurIPS).  NeurIPS, 2021.

---

> > > > > > ### Author Response · Authors · 2022-08-08
> > > > > > **Looking forward to your valuable feedback**
> > > > > >
> > > > > > Dear Reviewer Ewj4,
> > > > > >
> > > > > > Thank you again for the valuable comments. As the discussion period is close to the end, we want to reach out to see whether you find our responses satisfactory.
> > > > > >
> > > > > > We would like to hear from you about any further feedback, which is very important for us to improve the paper, and we really appreciate your time. If you still have any concerns, please let us know, and we would be very happy to respond.
> > > > > >
> > > > > > Best,
> > > > > >
> > > > > > Paper5481 Authors

---

### Official Review · Reviewer_47fC · 2022-07-08

**Rating:** 6
**Confidence:** 4
**Soundness:** 3 good
**Presentation:** 3 good
**Contribution:** 3 good

**Summary:**

This paper proposes a poisoning attack that amplifies the membership inference risk of an ML model on the samples from a specific class. The authors consider two attacks: dirty-label (label flipping of clean training samples) and clean-label (perturbing clean training samples). Essentially, these attacks aim to amplify the influence of member samples of the targeted class which allows for easier inference attacks.

**Questions:**

1) Do you measure the test accuracy only on the classes other than the target class t? Seems to me that the proposed attacks will hurt the accuracy on the target class significantly, which will no longer be stealthy.

**Limitations:**

- Consider running experiments on datasets where a specific class might be more sensitive to privacy leakage (for example, a medical dataset or a demographics dataset). It is difficult to understand why class targeted attack matters if the only experiments are on CIFAR.

- Consider evaluating some standard defensive strategies from prior work against poisoning.



**Strengths And Weaknesses:**

Strengths:

+ The synergy between poisoning and privacy leakage is interesting with profound implications and the proposed attacks are straightforward and effective.

+ Detailed ablation study to further understand the attack properties.

Weaknesses:

- It it difficult to say [34] is concurrent as it was on ArXiv for almost two months before NeurIPS deadline (and also from very visible researchers).

- The considered membership inference attack is somewhat weak, a more recent SOTA [Membership Inference Attacks From First Principles] might reduce the effectiveness of poisoning.

- Targeted attack is stealthy in the input space but many OOD detection algorithms nowadays operate on the latent space (which might be able to detect these attacks).

- No defenses considered against the proposed attack (outlier removal / differential privacy / regularization / strong data augmentation etc.)

---

> ### Author Response · Authors · 2022-08-02
> **Response to Reviewer 47fC**
>
> We thank the reviewer for the valuable comments (C). We hope that our responses (R) have fully addressed all of the reviewer’s concerns, and remain committed to clarifying any further questions that may arise during the discussion period.
>
> ***C1: It is difficult to say [34] is concurrent as it was on ArXiv for almost two months before NeurIPS deadline (and also from very visible researchers).***
>
> ***R1:***
> We are sure our work is original and should be considered concurrent to [34]. Please see the comment **Concurrent work** by reviewer gUo3 and our additional comments after it. We are fully committed to clarifying any further questions that may arise during the discussion period.
>
>
> ***C2: The considered membership inference attack is somewhat weak…***
>
> ***R2:***
> Thanks for your comments. The mentioned SOTA attack is based on a stronger attacker. They must obtain 1) sufficient shadow data and 2) train a large number of shadow models, which induces a high computation cost.
>
> Instead, we investigate the membership exposure with the evaluation protocol proposed by [1*, 2*]. Our attack assumes a weak attacker and has much lower computation costs and is more practical.
>
> We have also performed some preliminary experiments with the SOTA mentioned [3* ]. In our experiment, we trained 128 InceptionV3 shadow models on the CIFAR-10 dataset. For each shadow model, we randomly select 50% samples from $\mathcal{D}\_{clean} \cup \mathcal{D}\_{test}$ (i.e., all the members and non-members in our setup). The MI attack results with the mentioned SOTA [3*] are reported below.
>
> |||Without Poisoning|||Dirty-Label Poisoning|||Clean-Label Poisoning||
> |------|------|------|------|------|------|------|------|------|------|
> ||**MI AUC**|**TPR@FPR=1%**|**Test Acc.**|**MI AUC**|**TPR@FPR=1%**|**Test Acc.**|**MI AUC**|**TPR@FPR=1%**|**Test Acc.**|
> |InceptionV3|0.713$\pm$0.056|0.71$\pm$0.52%|0.676|0.799$\pm$0.053|0.94$\pm$0.60%|0.648$\pm$0.005|0.756$\pm$0.054|0.92$\pm$0.61%|0.663$\pm$0.002|
>
> We can see that our poisoning attack can help to improve the performance of the SOTA MI method [3*].
>
> (_Note: The experiment setup may affect the MI attack performance. In our case, each shadow model has been trained on about 10,000 samples in 20 epochs, with the feature extractor fixed. We use smaller shadow datasets and fewer shadow models than [3*], and we mainly consider the transfer learning setup, which may cause a weaker attack performance than that reported by [3*]. To achieve a stronger MI attack, more computation resources, and auxiliary data may be needed._)
>
> We would also like to emphasize that our goal is to show the potential of increasing membership exposure to data poisoning attacks. Our attack does not intend to further improve the MIA performance of SOTA methods.
>
> ***C3: Targeted attack is stealthy in the input space but many OOD detection algorithms nowadays operate on the latent space (which might be able to detect these attacks).***
>
> ***R3:***
> Thanks for your comments. Our targeted attacks aim at attacking the generic transfer learning pipeline, which has been demonstrated by many official tutorials of famous deep learning frameworks, such as [Tensorflow](https://www.tensorflow.org/tutorials/images/transfer_learning) and [PyTorch](https://pytorch.org/tutorials/beginner/transfer_learning_tutorial.html). However, these documents rarely warn the potential threats like ours or existing stealthy poisoning attacks against transfer learning [4*, 5*]. We hope our attacks
> Together with existing attacks can raise awareness and draw the ML practitioners’ attention to the security and privacy issues of the transfer learning pipeline.
>
> We will discuss your suggested OOD detection algorithms that can serve as potential countermeasures against attacks on the latent space in the revision.
>
> (cont'd)

---

> > ### Author Response · Authors · 2022-08-02
> > **Response to Reviewer 47fC (cont'd)**
> >
> > ***C4: No defenses considered against the proposed attack (outlier removal / differential privacy / regularization / strong data augmentation etc.)***
> >
> > *****R4:*****
> >  Thanks for your comments. We agree that designing defensive strategies is important and necessary. Indeed, we have evaluated the defense performance of differential privacy (DP) in Section C of the submitted appendix. The results show that DP can effectively prevent the student model from learning the poisons and hence, defend our proposed attack. Besides, we have also evaluated additional standard defense strategies against our dirty-label and clean-label attacks. Here we consider early stopping and regularization, and the results on the CIFAR-10 dataset are as follows.
> >
> > **Early stopping:** during the training process, for each epoch, we randomly sample out 10% of the training data as validation data, and use 90% of other data as training data. We monitor the validation loss, if it does not decrease in three epochs, we stop the training process. The results are as follows:
> >
> > |||Without Poisoning|||Dirty-Label Poisoning|||Clean-Label Poisoning||
> > |------|------|------|------|------|------|------|------|------|------|
> > ||**MI AUC**|**TPR@FPR=1%**|**Test Acc.**|**MI AUC**|**TPR@FPR=1%**|**Test Acc.**|**MI AUC**|**TPR@FPR=1%**|**Test Acc.**|
> > |Xception|0.568$\pm$0.043|1.00$\pm$0.44%|0.699|0.577$\pm$0.032|1.19$\pm$0.37%|0.769$\pm$0.000|0.576$\pm$0.032|1.18$\pm$0.36%|0.769$\pm$0.000|
> > |InceptionV3|0.598$\pm$0.046|0.95$\pm$0.23%|0.623|0.614$\pm$0.034|1.17$\pm$0.37%|0.675$\pm$0.001|0.621$\pm$0.032|1.09$\pm$0.30%|0.676$\pm$0.003|
> > |VGG16|0.561$\pm$0.036|1.12$\pm$0.44%|0.728|0.570$\pm$0.030|1.22$\pm$0.47%|0.798$\pm$0.003|0.572$\pm$0.030|1.29$\pm$0.38%|0.798$\pm$0.001|
> > |ResNet50|0.562$\pm$0.037|0.88$\pm$0.26%|0.772|0.567$\pm$0.026|1.07$\pm$0.34%|0.850$\pm$0.002|0.570$\pm$0.025|1.08$\pm$0.31%|0.851$\pm$0.001|
> > |MobileNetV2|0.561$\pm$0.037|1.40$\pm$0.48%|0.758|0.561$\pm$0.027|1.33$\pm$0.58%|0.840$\pm$0.000|0.561$\pm$0.027|1.40$\pm$0.54%|0.840$\pm$0.000|
> >
> > **Regularization:** For the fully connected layers, we add an L2-norm regularizer with a penalty of 0.05. The results are as follows:
> >
> > |||Without Poisoning|||Dirty-Label Poisoning|||Clean-Label Poisoning||
> > |------|------|------|------|------|------|------|------|------|------|
> > ||**MI AUC**|**TPR@FPR=1%**|**Test Acc.**|**MI AUC**|**TPR@FPR=1%**|**Test Acc.**|**MI AUC**|**TPR@FPR=1%**|**Test Acc.**|
> > |Xception|0.513$\pm$0.011|1.21$\pm$0.43%|0.722|0.514$\pm$0.014|1.31$\pm$0.67%|0.702$\pm$0.009|0.515$\pm$0.014|1.29$\pm$0.83%|0.702$\pm$0.006|
> > |InceptionV3|0.525$\pm$0.012|1.01$\pm$0.42%|0.643|0.523$\pm$0.011|1.09$\pm$0.56%|0.639$\pm$0.007|0.525$\pm$0.009|1.01$\pm$0.41%|0.643$\pm$0.006|
> > |VGG16|0.524$\pm$0.012|1.34$\pm$0.50%|0.793|0.526$\pm$0.010|1.48$\pm$0.60%|0.804$\pm$0.003|0.539$\pm$0.013|1.57$\pm$0.55%|0.797$\pm$0.002|
> > |ResNet50|0.519$\pm$0.009|1.22$\pm$0.52%|0.828|0.521$\pm$0.011|1.18$\pm$0.21%|0.807$\pm$0.009|0.528$\pm$0.011|1.41$\pm$0.48%|0.801$\pm$0.009|
> > |MobileNetV2|0.518$\pm$0.010|1.32$\pm$0.56%|0.798|0.523$\pm$0.008|1.12$\pm$0.32%|0.802$\pm$0.009|0.525$\pm$0.008|1.11$\pm$0.33%|0.798$\pm$0.008|
> >
> > The above results show that standard defensive strategies can help to defend our proposed attacks. We will include our discussion, evaluation results, and experiment codes about countermeasures in our revised paper and supplemental materials.
> >
> > (cont'd)

---

> > > ### Author Response · Authors · 2022-08-02
> > > **Response to Reviewer 47fC (cont'd)**
> > >
> > > ***C5: Do you measure the test accuracy only on the classes other than...***
> > >
> > > ***R5:***
> > > Our attacks will decrease the target class's accuracy. This is in line with most existing findings that have shown a strong connection between membership exposure and overfitting [6*, 7*]. It is inevitable to induce more overfitting (and consequently hurt the testing accuracy) to increase membership exposures. The trade-off exists by design. We would like to clarify that our attacks can still be considered to be stealthy from two aspects:
> > > - **Appearance:** For our clean-label poisoning attack, we make the perturbations as invisible as possible and keep the original label unchanged. As a result, these poisons look normal to human moderators or labelers. It is hard to identify our attack if the victim does not implement outlier detectors in the feature space.
> > > - **Model Performance:**  As discussed above, decreasing the testing accuracy on the target class is inevitable. But the subclass accuracy degradation would only be observed when the model developer examines the model performance carefully. Suppose a layman runs the *evaluate()* function on the whole testing dataset. They would likely obtain satisfactory testing accuracy without noticing their model has been poisoned. In addition, even observing the decreased accuracy on the target class, it is still a bit hard for the model developer to find out the outliers (i.e., poisons) as they have a natural appearance. Actually, there exists a game between the attacker and the defender. We have shown an example of dirty-label attacks against InceptionV3-based CIFAR-10 classifiers. The attacker can sacrifice some MIA performance while not having a discernible accuracy drop. For instance, the attacker can choose the poisoning budget $b_{poison}=200$, to increase the average AUC from 0.733 to 0.817, with testing accuracy decreasing from 67.7% to 62.5%.
> > >
> > > |$b_{poison}$|1000|800|600|400|200|100|50|0|
> > > |------|------|------|------|------|------|------|------|------|
> > > |**Test Acc. on the target class**|0.518$\pm$0.048|0.526$\pm$0.046|0.550$\pm$0.061|0.595$\pm$0.067|0.625$\pm$0.073|0.646$\pm$0.069|0.658$\pm$0.079|0.677$\pm$0.078|
> > > |**MI AUC on the target class**|0.935$\pm$0.016|0.922$\pm$0.013|0.901$\pm$0.019|0.866$\pm$0.026|0.817$\pm$0.037|0.777$\pm$0.044|0.756$\pm$0.052|0.733$\pm$0.058|
> > > |**TPR@FPR=1% on the target class**|7.31$\pm$2.83%|4.11$\pm$1.25%|3.02$\pm$1.26%|2.13$\pm$0.58%|1.27$\pm$0.48%|1.13$\pm$0.40%|1.00$\pm$0.29%|1.34$\pm$0.47%|
> > >
> > > We believe our proposed attacks, as well as prior clean-label poisoning attacks, have posed realistic threats to the generic transfer learning pipeline.
> > >
> > > ***C6: Consider running experiments on datasets where a specific class might be more sensitive to privacy leakage...***
> > >
> > > ***R6:***
> > > Thanks for your comments, and we agree that it is important to conduct evaluations on privacy-critical datasets.
> > >
> > > In our submitted paper, despite three benchmark datasets (MNIST, CIFAR-10, and STL-10), we have also evaluated our attacks on two privacy-critical datasets, PatchCamelyon and CelebA. The former consists of medical images to predict the presence of metastatic tissues, and the latter consists of facial images. We report the attack results in Table 1, and we show some attack examples in Section A of the appendix. Evaluation results show that our attacks are effective in improving the membership inference chance on both medical and facial datasets.
> > >
> > > ***C7: Consider evaluating some standard defensive strategies from prior work against poisoning.***
> > >
> > > ***R7:***
> > > We’ve evaluated more defense strategies. Please refer to R4 for more details.
> > >
> > > **References**
> > >
> > > [1*] Liwei Song et al. Privacy Risks of Securing Machine Learning Models against Adversarial Examples. In ACM SIGSAC Conference on Computer and Communications Security (CCS), pages 241–257. ACM, 2019.
> > >
> > > [2*] Liwei Song et al. Systematic Evaluation of Privacy Risks of Machine Learning Models. In USENIX Security Symposium (USENIX Security), pages 2615–2632. USENIX, 2021.
> > >
> > > [3*] Florian Tramèr et al. Truth Serum: Poisoning Machine Learning Models to Reveal Their Secrets. arXiv 2204.00032, 2022.
> > >
> > > [4*] Ali Shafahi et al. Poison Frogs! Targeted Clean-Label Poisoning Attacks on Neural Networks. In Annual Conference on Neural Information Processing Systems (NeurIPS), pages 6103–6113. NeurIPS, 2018.
> > >
> > > [5*] Chen Zhu et al. Transferable Clean-Label Poisoning Attacks on Deep Neural Nets. In International Conference on Machine Learning (ICML), pages 7614–7623. PMLR, 2019.
> > >
> > > [6*] Reza Shokri et al. Membership Inference Attacks Against Machine Learning Models. In IEEE Symposium on Security and Privacy (S&P), pages 3–18. IEEE, 2017.
> > >
> > > [7*] Ahmed Salem, Yang Zhang, Mathias Humbert, Pascal Berrang, Mario Fritz, and Michael Backes. ML-Leaks: Model and Data Independent Membership Inference Attacks and Defenses on Machine Learning Models. In Network and Distributed System Security Symposium (NDSS). Internet Society, 2019.

---

> > > > ### Comment · Reviewer_47fC · 2022-08-07
> > > > **Thank you for your responses**
> > > >
> > > > I read your responses and I appreciate the nuance they bring to your paper. Especially about the defenses, you show the results I expected (that simple countermeasures will defeat your attack). This doesn't mean that your attack is weak or useless. Adding these discussions to your paper and tuning down your claims (e.g., your attack is undetectable) will make your paper much stronger and valuable to the community.
> > > >
> > > > I will increase my score and I hope you will incorporate these discussions into your paper (not just into the Appendix but to the main paper as well). The overall lessons is that, it is possible to detect/defeat such attacks but the standard implementations do not include the necessary checks/countermeasures (e.g., early stopping, strong regularization, evaluating accuracy for each class individually etc.).
> > > >
> > > > It is also a good idea to use the metrics in [1,2] to evaluate the MI success, rather than the simple MI attacks. Please consider supplementing the results in the main paper with these metrics.

---

> > > > > ### Author Response · Authors · 2022-08-07
> > > > > **Thank you for your valuable comments**
> > > > >
> > > > > Thank you again for your valuable comments to help improve the paper!
> > > > >
> > > > > We will add the suggested discussions on defenses to the paper both in the main paper and appendix. Meanwhile, we will also evaluate our attacks with more MI attack metrics.

---

### Review · Ethics_Reviewer_nWWT · 2022-08-05

**Recommendation:**

I think it would be sufficient to bring the results on defenses against the propose attacks and some discussion to the forefront. First, the potential negative impact is not stressed in the main paper (although obvious, it is still important to state it). Second, the current version claims that defenses are left as future work. Instead I would suggest bringing the DP mitigation strategy to the main body and make it part of the main story of the paper (include it in the abstract etc). I would also suggest adding the other two approaches although I am reluctant to do that because technical reviewers might not be able to verify correctness at this stage of the reviewing process.

I believe that with these two changes the paper could address the ethical concerns in the current version.

**Ethical Issues:**

Yes

**Ethics Review:**

This paper proposes two strategies for data poisoning which are shown to amplify the success of membership inference attacks. That is, via inserting poisoned datapoints in the training dataset, the chance of identifying members from a specific targeted class from the training sample increases. This work has obvious potential negative impact as it describes techniques which if employed by attackers would incur privacy risks for ML models.

---

### Review · Ethics_Reviewer_1fDd · 2022-08-06

**Recommendation:**

Yes, my recommendation is that the authors add a discussion of the questions highlighted above in the main paper.  Further, the technical defense presented in the appendix should be at least mentioned in the main paper if not integrated into it.

**Ethical Issues:**

Yes

**Ethics Review:**

This paper demonstrates the use of a data poisoning attack that exacerbates membership inference attacks (i.e., information leakage or potential privacy attack)

This is an important and interesting demonstration.  But there are also important ethical questions to consider in this research.  These questions should be discussed, and the authors' thoughts and decisions clearly stated:

1. Primarily, what is the potential benefit and damage that can be done by making people aware of this attack?
    * The additional material on differential privacy as a defense against this poisoning and leakage attack is great to see.  It would be good to see this integrated into the main paper.
    * Are there any critical applications or actors that are more or most at risk from this attack?  Is there specific guidance for them?  Have they been approached and notified of this attack already?
    * DP here is a protective mechanism that can be implemented by the ML system.  Is there a defense for individuals who wish to protect their own privacy?
2. Do all details of the attack need to be published to achieve the benefits of making people aware of this attack?  E.g., are there key details that could have been withheld, or is that infeasible?
3. Are there further implications of this work that should be considered?

---

### Comment · Reviewer_gUo3 · 2022-07-26
**Concurrent work**

I'd like to challenge the criticism from reviewers 47fc and Ewj4 that the very similar work of Tramer et al. [34] shouldn't be considered as concurrent work by the authors of this paper.

The Tramer et al. paper appeared on arxiv on March 31st 2022 (https://arxiv.org/abs/2204.00032).
This is 6 weeks before the NeurIPS abstract deadline, and 7 weeks before the submission deadline.

It is inconceivable to me that authors should be expected to quantitatively compare to, or otherwise improve upon, a paper that was clearly being written *concurrently* to this NeurIPS submission (unless we expect people to write their NeurIPS papers in <6 weeks...)

---

> ### Author Response · Authors · 2022-08-02
> **Response on the concurrent work**
>
> Thanks for the comments.
>
> In addition to the comments by Reviewer gUo3, we would like to emphasize the differences between our attack with the concurrent work of Tramèr et al. [1*].
>
> - Our proposed attack mainly aims at the *untargeted attack*. Tramèr et al. [1*] mainly aims at the *targeted attack* and requires the attack to determine which sample $(x,y)$ to infer in advance.
> - Besides the classic label-flipping poisoning attack [2*], we also investigate the feasibility of clean-label poisoning. With clean-label poisoning attacks, the attacker can craft more natural and stealthy poisoning samples. We believe that our work can be a good complement to existing clean-label poisoning attacks that try to cause misclassification.
>
> We strongly agree that novelty and originality are very important for NeurIPS, and we are sure that our work is original.
>
> We hope that our responses have sufficiently addressed the reviewer’s concerns about the originality. We are fully committed to clarifying any further questions that may arise during the discussion period.
>
> **References**
>
> [1*] Florian Tramèr, Reza Shokri, Ayrton San Joaquin, Hoang Le, Matthew Jagielski, Sanghyun Hong, and Nicholas Carlini. Truth Serum: Poisoning Machine Learning Models to Reveal Their Secrets. arXiv 2204.00032, 2022.
>
> [2*] Battista Biggio, Blaine Nelson, and Pavel Laskov. Poisoning Attacks against Support Vector Machines. In International Conference on Machine Learning (ICML). icml.cc / Omnipress, 2012.

---

### Author Response · Authors · 2022-08-02
**Typo in Equation (3)**

We are sincerely sorry for the typo in Equation (3), which might cause misunderstandings.

The constraint should be

$$
\text{s.t.} \|x’-x\|_{\infty} \leq \epsilon, x’ \in \mathcal{X}
$$

And accordingly, $x_{min}$ and $x_{max}$ in Line 206 should be

$$
x_{min}=\text{max}(0, x-\epsilon), x_{max}=\text{min}(1, x-\epsilon)
$$

By Equation (3), we have two goals:

- We would like the poisoning sample $(x^*,y)$ to have a similar appearance to the sample $(x,y)$ (so we constrain $\|x^*-x\|_{\infty} \leq \epsilon$).

- Meanwhile, we hope the poisoning sample to have nearly the same latent feature with a base sample $(x_{base}, t)$ (so we aim to make $g(x^*) \approx g(x_{base})$).

We will correct the typo in the revision. We are committed to clarifying any further questions that may arise during the discussion period if there is any confusion raised by the typo.

---

### Author Response · Authors · 2022-08-07
**Looking forward to further feedbacks**

Dear Reviewers,

Thank you again for your thoughtful comments and constructive suggestions. We have posted our responses to the detailed comments.

We totally understand that you are really busy at this time, as you have to respond to the rebuttals of multiple assigned papers.

We would deeply appreciate your feedback on whether the points made by the reviewers have been addressed. If there are any remaining issues, we will try our best to address them.

Best,

Paper5481 Authors

---

### Meta-Review · Area_Chair_pmLq · 2022-08-24

**Recommendation:** Accept
**Confidence:** Certain

**Metareview:**

Reviewers 47fC and gUo3, who are experts in this area, were very pleased by the author discussion and clarification. They appreciated that the authors will tone down their claims and reduce confusion in the final version. As gUo3 said, there are concurrent works, but that can't hurt this work -- it only shows that there is significant interest in this topic. The same reviewer did feel that the scope of the results seem somewhat weak (just transfer learning), but there is some evidence that things extend beyond that setting. If the authors wish to further improve the work, it may be worth performing further experimental evaluation.

Note that the two other reviewers, who were negative, neither engaged with the authors nor myself when prompted to in private deliberation. Since they appear unwilling to defend their negative evaluation of the paper, I defer to the positive sentiments of the other two reviewers who did discuss privately.

**Award:**

No

---

### Decision · Program_Chairs · 2022-09-14

Accept